



# Evaluation of a cloudy cold-air pool in the Columbia River Basin in different versions of the HRRR model

Bianca Adler[1,2], James M. Wilczak[2], Jaymes Kenyon[3,a], Laura Bianco[1,2], Irina V. Djalalova[1,2], Joseph B. Olson[3], and David D. Turner[3]

[1]CIRES, University of Colorado, Boulder, CO, USA
[2]NOAA Physical Sciences Laboratory, Boulder, CO, USA
[3]NOAA Global Systems Laboratory, Boulder, CO, USA
[a]Current affiliation: NOAA National Weather Service, Grand Rapids, MI, USA

**Correspondence:** Bianca Adler (bianca.adler@noaa.gov)

**Abstract.** The accurate forecast of persistent orographic cold-air pools in numerical weather prediction models is essential for the optimal integration of wind energy into the electrical grid during these events. Model development efforts during the Second Wind Forecast Improvement Project (WFIP2) aimed to address the challenges also related to this. We evaluated three different versions of NOAA's High-Resolution Rapid Refresh model with two different horizontal grid spacings against in

situ and remote sensing observations to investigate how developments in physical parameterizations and numerical methods targeted during WFIP2 impacted the simulation of a persistent cold-air pool in the Columbia River Basin. Differences between the different model versions were in particular visible in the simulated temperature and low-level cloud fields. The model developments led to an enhanced low-level cloud cover in the cold pool, resulting in better agreement with the observations. This removed a diurnal cycle in the near-surface temperature bias at stations throughout the basin by reducing a cold bias

during the night and a warm bias during the day. However, low-level clouds did not clear sufficiently during daytime in the newest model version, which led to a warm bias near-the surface during the second night of the reforecasts and leaves room for further model developments. Besides the improvements during the persistent phase of the cold pool, the model developments also led to a better representation of its decay by slowing down its erosion.

## 1 Introduction

Persistent cold-air pools (hereafter *cold pools*) frequently form during wintertime in orographic basins and valleys, when the air near the surface cools and/or the air aloft warms which leads to an increase of temperature with height (e.g. Whiteman et al., 2001). Wind speed within the cold pools is generally weak and rapidly accelerates during its decay. Since many wind turbines have a cut-in speed of around 3 m s$^{-1}$, the wind energy produced during cold pool events is typically small (e.g. Bianco et al., 2016; Wilczak et al., 2019). At the end of the events, wind energy production often jumps to very large values, so called wind

energy ramps, because wind power increases approximately as the cube of the wind speed. Forecast errors in wind speed and cold pool decay time were found to be reduced by an improved representation of the cold pool's vertical structure and depth (e.g. Olson et al., 2019b). To more efficiently integrate the wind energy produced during cold pool events into the electrical





grid, accurate forecasts of the evolution and structure of these events are necessary (e.g. Wilczak et al., 2019; Olson et al., 2019b). Cold pools are complex and their evolution and structure depends on a variety of atmospheric processes including

large-scale advection and subsidence, mesoscale flows as well as radiative, turbulent and cloud processes (e.g. Lareau et al., 2013). This makes it challenging for numerical weather prediction models to correctly represent the cold pool structure and evolution (e.g. Reeves et al., 2011; Holtslag et al., 2013; Olson et al., 2019b). Numerical simulations indicate that the relative importance of the different processes varies from one cold-air pool to another, and may depend on the geographic location, terrain characteristics, season of the year, large-scale conditions, and snow cover (e.g. Zhong et al., 2001; Wei et al., 2013; Lu

and Zhong, 2014; Neemann et al., 2015; Lareau and Horel, 2015; Crosman and Horel, 2017).

Low-level clouds can have a strong impact on the temporal evolution and vertical structure of temperature in cold pools. While cloud-free cold pools usually have a surface-based temperature inversion in which stability decreases smoothly with height, cloudy cold pools often have a near-moist adiabatic lapse rate in the sub-cloud and cloud layer which is topped by a strong elevated temperature inversion (Whiteman et al., 2001). Strong longwave radiative cooling at cloud top causes the

elevated temperature inversion and can induce top-down turbulent mixing (e.g. Adler et al., 2021). Increased mixing near the surface in cloudy cold pools was found to be associated with lower pollutant concentrations compared to cloud-free conditions (e.g. VanReken et al., 2017). The composition of the low-level clouds, whether ice-dominant or liquid-dominant clouds, was found to be relevant for the vertical structure of simulated wintertime cold pools (Neemann et al., 2015). When ice-phase particles dominated in the simulated cloud layers, the cloudy layer was shallower and temperature near the surface was lower

compared to liquid-dominant clouds. These authors found that ice-dominant clouds gave the best agreement between observed and simulated temperature profiles and cloud occurrences. In the presence of clouds, the diurnal temperature cycle near the surface is weakened due to reduced shortwave downward radiation flux during daytime and reduced longwave net radiation flux during nighttime (e.g. Sun and Holmes, 2019). Too few simulated nocturnal low-level clouds can lead to large cold biases in the simulated near-surface temperature during nights when low-level clouds are observed as they reduce longwave outgoing

radiation flux (e.g. Hughes et al., 2015) and increase downward longwave radiation flux below the clouds. On the other hand, daytime low-level clouds can help to maintain the cold pool as they reduce convective heating (e.g. Zhong et al., 2001). This is especially relevant during spring and fall when insolation is strong. The failure of models to produce realistic low-level clouds can thus lead to an erroneous erosion of the cold pool during daytime under certain conditions.

To address the challenges numerical weather prediction models have to correctly forecast wind over complex terrain and

to improve the forecast for wind energy applications, the Second Wind Forecast Improvement Project (WFIP2) was initiated by the Department of Energy in 2015 (Shaw et al., 2019). The target area was the Columbia River Basin in Washington and Oregon in the Pacific Northwest of the United States, which is home to a large amount of wind energy production (more than 6 GW at the time of the project) and is often affected by cold pools (McCaffrey et al., 2019), gap flows (Neiman et al., 2019), and mountain waves (Draxl et al., 2021) - atmospheric phenomena which are all relevant for wind energy forecasts (Wilczak et al.,

2019). Besides the conduction of a comprehensive 18-month long field campaign (Wilczak et al., 2019) and the development of support tools to assist the industry in wind power forecasting, model development efforts were a key component of WFIP2 (Olson et al., 2019b). The basis for the model developments were NOAA's Rapid Refresh (RAP, Benjamin et al. (2016)) and



High-Resolution Rapid Refresh (HRRR, Dowell et al. (2022)) models. In this study, we used three different versions of the HRRR model to evaluate how the model developments impacted the characteristics of a strong and persistent cold pool event
in the Columbia River Basin, which occurred within a 10-day period from 10-19 January 2017.

The observed evolution and structure of this cold pool event, as well as the involved processes, were investigated in detail by Adler et al. (2021) using the comprehensive observational data set gathered during the WFIP2 field campaign together with surface measurements from several hundred surface stations available from the MesoWest repository (Horel et al., 2002). Adler et al. (2021) found that the cold pool structure was strongly modulated by the presence of low-level clouds and that its temporal
evolution was largely driven by synoptic-scale processes; that is, horizontal advection and subsidence played an important role during its formation and maintenance. The cold pool decayed when a low-pressure system approached the area and warm air gradually descended into the basin decreasing the cold pool depth. The observations indicated that strong downslope winds formed on the southern slopes of the basin during the decay. The same cold pool event was chosen for two numerical studies to address the impact of modified physical parameterizations. Arthur et al. (2022) investigated the effects of different planetary
boundary layer (PBL) schemes and horizontal diffusion treatment on the representation of the cold pool. These authors found that computing horizontal diffusion in physical space and using a three-dimensional PBL scheme better represented the vertical wind and temperature structure in the cold pool and its decay compared to a standard one-dimensional PBL scheme. Berg et al. (2021) examined the sensitivity of wind speed at turbine hub height to parameters in the Mellor-Yamada-Nakanishi-Niino eddy-diffusivity mass-flux scheme (MYNN-EDMF) parameterization using the same cold pool event as an example for a
winter period. They found a high bias in hub height wind speed which was fairly insensitive to changes in the parameter values and concluded that the representation of cold pool dynamics cannot be improved with changes in the MYNN-EDMF parameterization. In addition, since the mass-flux component of the EDMF parameterization was rarely triggered during the cold pool event, there was little sensitivity to these parameters.

In this study, we first evaluate how the model developments targeted during WFIP2 impact the cold pool characteristics
during its maintenance and decay phase by comparing 24-h reforecasts for the different model versions to observations (Sect. 3). This includes the comparison of simulated and observed temperature and wind speed profiles (Sect. 3.1) as well as the cold pool strength (Sect. 3.2) computed from ground-based remote sensing instruments. Using a large number of surface stations distributed in the basin, we further investigated the dependence of the near-surface temperature bias on station height and related differences between the model versions to differences in low-level cloud cover using downward shortwave and longwave
radiation fluxes as proxies (Sect. 3.3). In a second step, we investigate how well the newest model version performed when extending the reforecast horizon to 48 h by comparing the first 24 h of the reforecasts to the last 24 h (Sect. 4). We started with assessing differences in temperature and cloud profiles (Sect. 4.1), followed by an investigation on how differences in the cold pool thermodynamic structure during its maintenance phase may impact its decay (Sect. 4.2). In a final step, we evaluated differences in the near-surface temperature bias and its relationship to clouds (Sect. 4.3).



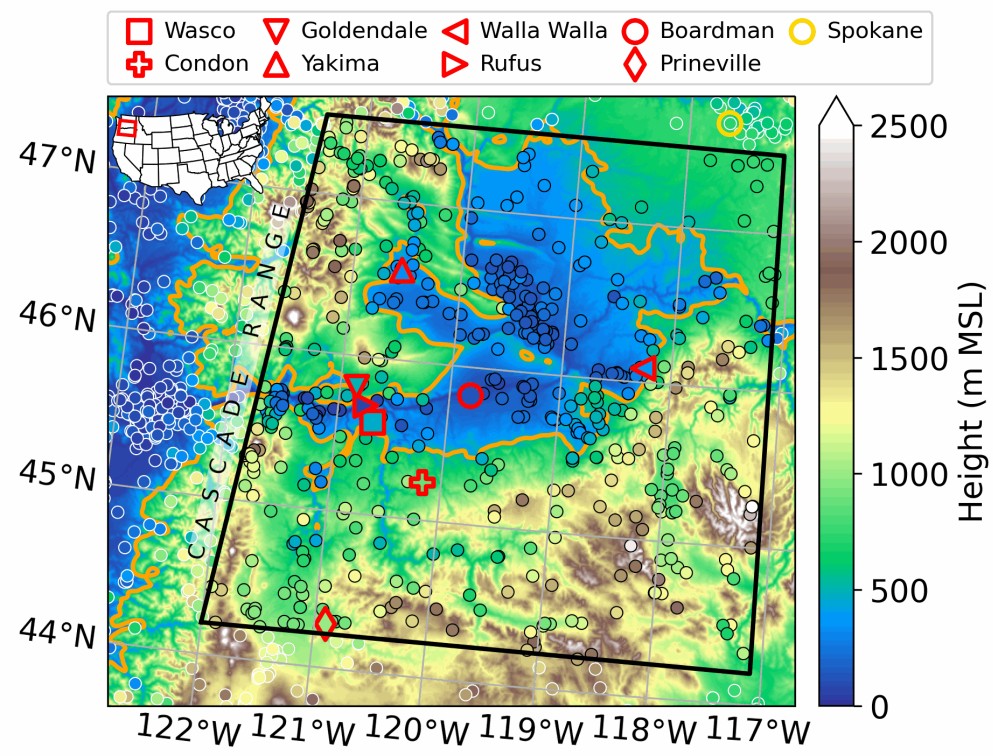

**Figure 1.** Terrain height in the investigation area and location of stations with profile and radiation flux measurements from the WFIP2 campaign (red markers) and from the MesoWest repository (black and white circles). The fill color of each marker indicates the station height. Stations within the black-outlined polygon, markers with red and black outline, are used for the model evaluation. The yellow circle indicates the location of the radiosonde station at Spokane. The 500 m terrain contour is given by the orange isoline. Terrain data are from the d02 runs with 750 m horizontal grid spacing. The inset plot in the upper-left corner shows the location of the investigation area in the United States.

## 2 Investigation area, observational data, and model configurations

The Columbia River Basin is a large basin in Washington and Oregon which is several hundred kilometers in diameter and is bordered by the north-south oriented Cascade Range to the west (Fig. 1). It has a broad northern part and an elongated and narrow southern part, which is well visible from the 500 m terrain contour line (orange contour in Fig. 1). Adler et al. (2021) estimated a mean ridge height of 1244 m MSL in the investigation area, which we adapted here.

### 2.1 Observational data

To evaluate the different model runs (Sect. 2.2), we compared the model output to temperature and wind speed profiles retrieved from ground-based remote sensing observations and in situ near-surface measurements. The different observational sources



**Table 1.** Station altitude and measurement types at WFIP2 sites.

| Site | Altitude (m MSL) | Measurement type |
|------|------------------|------------------|
| Rufus | 62 | Radiation |
| Boardman | 112 | Radiation, wind profile |
| Yakima | 329 | Wind profile |
| Walla Walla | 381 | Wind profile |
| Wasco | 456 | Radiation, wind profile, temperature profile, cloud base height |
| Goldendale | 502 | Wind profile |
| Condon | 891 | Radiation, wind profile |
| Prineville | 991 | Radiation, wind profile |

were the same as used by Adler et al. (2021) and we refer the reader to that manuscript for a detailed description of the instruments and methods.

The WFIP2 instrumentation was concentrated in the southern part of the basin with the site at Wasco being the best equipped (Wilczak et al., 2019). Locations of WFIP2 sites are indicated by red markers in Fig. 1 and information on station altitude and measurement types at the different sites are given in Tab. 1. Seven radar wind profilers were deployed in the basin for WFIP2. They operated at 915-MHz and provided hourly averaged horizontal wind profiles roughly between around 100 and 2000 m above ground. At Wasco, a microwave radiometer was collocated with a radio acoustic sounding system (RASS) associated
with the radar wind profiler. The brightness temperatures measured by the microwave radiometer were combined with the virtual temperature profiles of the RASS and near-surface measurements of temperature and humidity using an optimal estimation physical retrieval (Tropospheric Remotely Observed Profiling via Optimal Estimation (TROPoe), Turner and Löhnert, 2014; Turner and Blumberg, 2019; Turner and Löhnert, 2021) to obtain the best possible information on thermodynamic profiles in the troposphere from these instruments (Djalalova et al., 2022). The output frequency of the TROPoe retrievals was
15 min. Detailed information on the application of the retrieval for the investigated period are given in Adler et al. (2021). Ceilometer measurements at Wasco provided cloud base height estimates every 16 s and 15-min averages of shortwave and longwave downward radiation fluxes were available at Rufus, Boardman, Wasco, Condon, and Prineville. Spatial information on shortwave downward radiation flux at the surface was obtained from the Geostationary Operational Environmental Satellites (GOES) Solar Insolation Product (GSIP) with a horizontal grid spacing of 4-km (e.g. Sengupta et al., 2014).

One-hour averages of near-surface temperature measurements were available from a large number of stations distributed in the investigation area (white- and black-outlined circles in Fig. 1). The data are available through MesoWest, which provides observations from many different organizations (Horel et al., 2002) and can be downloaded with the Mesonet API (Synoptic Data, 2021). The applied quality checks are detailed in Adler et al. (2021). For the model evaluation, more than 500 stations located in the Columbia River Basin and the surrounding higher terrain are used (circles with black outlines within the black
polygon in Fig. 1). Approximately 200 of these stations are located in the lower part of the basin with station altitudes of less than 500 m MSL, around 200 are between 500 and 1250 m MSL, and nearly 100 stations are located above 1250 m MSL.



| HRRR version | Domain | Forecast time | Abbreviation | Comment |
|---|---|---|---|---|
| CTL | d01 | | CTL d01 | Pre-WFIP2 physics package (≈ HRRR version 1) |
| | d02 | | CTL d02 | |
| EXP | d01 | | EXP d01 | Physics improvements developed during WFIP2 |
| | d02 | | EXP d02 | |
| v4 | d01 | | v4fp1 d01 | Further physics modifications (≈ HRRR version 4 except for wind farm parameterization) |
| | d02 | | v4fp1 d02 | |
| | d01 | | v4fp2 d01 | |
| | d02 | | v4fp2 d02 | |

**Figure 2.** Overview of the eight model runs used in this study. Three versions of HRRR (CTL, EXP, and v4) are run for two domains with different horizontal grid spacing (d01: $\Delta x$=3 km and d02: $\Delta x$=750 m). CTL and EXP runs are 24 h forecasts and the v4 runs are 48 h forecasts. d01 runs are initialized every 24 h at 0000 UTC and nested d02 runs are initialized 3 h later from the 3-h d01 forecast.

## 2.2 Model configurations

Three different versions of the HRRR model and two horizontal grid spacings were evaluated in this study. An overview of the different versions, forecast periods, and abbreviations is given in Fig. 2. The first model version is the WFIP2 control version (hereafter *CTL*) and corresponds to the configuration of HRRR version 1, which was run operationally at the beginning of WFIP2. The second model version (*EXP*) uses model developments in physical parameterizations and numerical methods which were targeted in WFIP2. The third model version (*v4*) used is very close to the currently operational HRRR version 4, which encompasses many of the changes present in EXP or refinements of those. A summary of the main model developments in EXP and v4 is given below. More details on WFIP2 specific model changes can be found in Olson et al. (2019b) and Olson et al. (2019a) and on general HRRR developments in Dowell et al. (2022) and James et al. (2022).

    – The mixing length in the MYNN PBL scheme was revised to improve the forecast performance in stable PBLs and to better maintain inversions. This was accomplished by allowing the mixing length to be independent of height above



ground when strong static stability is present and reducing the magnitude of the buoyancy length scale, which is the primary limiting length scale in stable conditions.

– A mass flux scheme was included in the MYNN PBL scheme making it an EDMF scheme. This new component represents the nonlocal turbulent transport by thermal plumes in convective PBLs and overall improves the coverage of shallow cumulus and profiles of temperature and humidity. However, during this cold pool event, the mass flux scheme was largely inactive as the surface sensible heat flux was too small to trigger it.

– A subgrid-scale scale cloud representation was implemented which improves the representation of subgrid-scale stratus
and shallow cumulus, and enables the interaction between these clouds and the simulated downward shortwave radiation flux which had a strong impact on the surface energy balance. There was no subgrid-scale cloud parameterization in the CTL run. Changes were made between the EXP and v4 version to allow more liquid water and smaller effective radii in the subgrid-scale clouds in the latter. James et al. (2022) attributed an overall reduction of shortwave downward radiation bias in v4 when averaging over the conus to improvements in the subgrid-scale cloud scheme.

– A small-scale gravity wave drag and a form drag due to subgrid-scale orography was added. The former is active only in stable PBLs and reduces the near-surface wind speed and the shear-generated turbulence, which improves the maintenance of cold pools. Both are only active when the horizontal grid spacing is smaller than 1 km.

– A wind farm parameterization was added.

– Horizontal diffusion along terrain-following $\sigma$ levels was replaced with diffusion in Cartesian space which improves the
maintenance of cold pools by no longer mixing vertically when $\sigma$ levels follow steep terrain.

– A 6th order filter was implemented which reduced filtering of scalars and momentum over steep terrain and helped to maintain clouds and cold pools. While this filter was already present in EXP, a lower diffusion parameter was used in v4 which shuts off mixing more aggressively.

Each model version was run for two domains with different horizontal grid spacing. The outer domain encompasses the
western United States ($\Delta x = 3$ km, hereafter referred to as *d01*). A nested domain was centered on the Columbia River Basin with $\Delta x = 750$ m (hereafter referred to as *d02*). The domains are shown in Olson et al. (2019b), their Fig. 2. For this study, we used d01 runs which were initialized every 24 h at 00:00 UTC. The model was run in a 'cold-start' configuration, where initial conditions were supplied from the RAP model without additional data assimilation or antecedent cycling (Olson et al., 2019b). After 3 h of spin-up during which the model atmosphere adjusted to the higher resolution terrain, the nested d02 runs
were initialized from the 3-h d01 forecast. The model output was written every 15 min. To be consistent for the different runs we only evaluated the model output for times larger than forecast hour 3. The CTL and EXP runs were not specifically carried out for this study, but were conducted as part of WFIP2 model development efforts (Olson et al., 2019b) and were run for 24 h. The investigated cold pool event fell within the winter reforecast period (Bianco et al. (2021), their Table 2). Unfortunately, no model output was stored for 18 Jan for both EXP runs and for the d01 CTL run. The v4 version was run specifically for this



study and the reforecast horizon was extended to 48 h, to allow an investigation of how the representation of the cold pool may have changed for longer forecast hours. For this purpose, we split the v4 runs into two forecast periods with forecast period 1 (v4fp1) encompassing hours 3-24 and forecast period 2 (v4fp2) encompassing hours 24-48. Note that we only used forecast hours 27-48 for fp2 when computing statistics in order to compare the same hours of the day for v4fp1 and v4fp2. To study clouds, additional output variables were added in v4 which included liquid water path (LWP) as well as profiles of cloud water,

snow, and ice mixing ratios. LWP and mixing ratios include contributions from the resolved (gridscale) and subgrid-scale clouds.

To compute the differences between the model and the observations, model variables were bilinear interpolated to the latitude/longitude of the measurement sites and temperature and wind simulated profiles were further linearly interpolated to the measurement heights. Temperature profile and radiation biases were defined as model data minus observation data

and computed every 15-min. Since wind speed profiles and near-surface temperature measurements were available as hourly averages, hourly averaged model data were used for the computation of these biases.

## 3 Evaluation of three different HRRR versions

In this section, different characteristics of the cold pool are evaluated for each of the CTL, EXP, and v4fp1 runs (details in Fig. 2). This includes the analysis of biases between the simulated and observed temperature and wind speed profiles

(Sect. 3.1), the temporal evolution of the cold pool strength (Sect. 3.2), and the dependence of the near-surface temperature biases on station height (Sect. 3.3).

### 3.1 Bias of temperature and wind speed profiles

Figures 3a,b illustrate the observed temperature and wind evolution at Wasco during the 10-day period. A detailed description of the temporal evolution of the cold pool and the relevant processes is given in Adler et al. (2021). During the first three

days, a strong snowstorm passed the area associated with a decrease in temperature and strong north-easterly flow, which was channeled along the west-east valley axis (Figs. 3a,b) and left the ground in the investigation area snow covered during the cold pool period. Towards the end of 12 January, warming above around 1200 m MSL, that is above the mean ridge height, initiated the formation of the cold pool, which was quickly enhanced by decreasing temperatures in lower layers (Fig. 3a). From 13 throughout 16 January (hereafter referred to as *CAP* period), the cold pool was maintained and experienced a layered structure:

adjacent to the surface, a layer of a few hundred meters depth exhibited a diurnal cycle due to daytime warming and nighttime radiative cooling. Above this layer up to approximately the mean ridge height, the temperature structure was near isothermal and showed no diurnal variability. This layer was topped by a strong elevated inversion up to around 2000 m MSL. Low-level clouds and fog were present during the CAP period, with cloud base heights usually less than 300 m above ground, and these impacted the low-level stratification. On 15 and 16 January when cloud base heights were fairly constant and persistent with

time, the sub-cloud layer was neutrally-stratified, while a more stable stratification existed on 13 and 14 January when cloud base heights were more variable. Wind within the cold pool was generally weak and from the easterly direction (Fig. 3b). With



**Figure 3.** Time height section of observed (a) temperature profiles (color-coded) and potential temperature (isolines) and (b) horizontal wind speed (color-coded) and horizontal wind vector (arrows) at Wasco. Time height section of (c,e,g) temperature bias ΔT and (d,f,h) horizontal wind speed bias ΔU (d,f,h) for d02 CTL, EXP, and v4fp1 runs at Wasco. The green dots show observed cloud base height in (a) and the green contours show the simulated 0.1 g kg$^{-1}$ isoline of total condensate (cloud water, snow and ice mixing ratio) in (g) (only for v4fp1). The horizontal grey line indicates the mean ridge height, dark grey shading station height, and light grey shading missing data. The CAP and Decay periods are indicated by the vertical dashed lines.





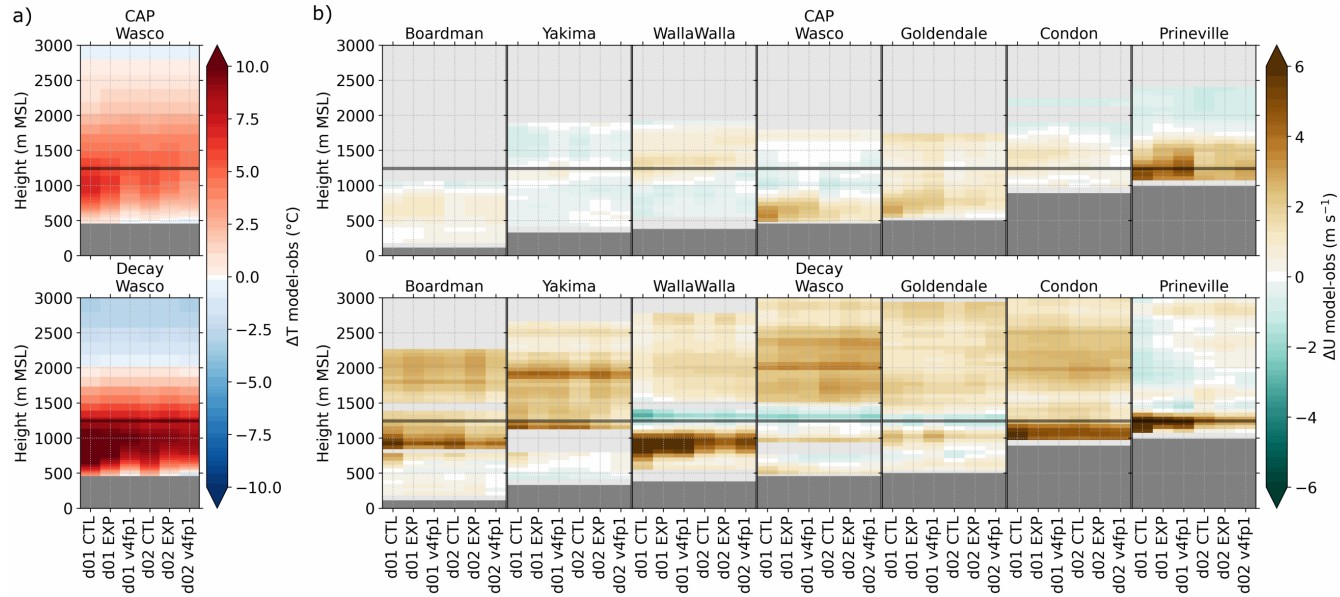

**Figure 4.** Profiles of mean temperature bias $\Delta T$ (a) and horizontal wind speed bias $\Delta U$ (b) averaged over the CAP and Decay periods for EXP, CTL, and v4fp1 runs at different locations in the Columbia River Basin. The horizontal grey line indicates the mean ridge height, dark grey shading station height, and light grey shading missing data.

the approach of a low-pressure system from the northwest, strong southwesterly wind and first warm and then cold air reached the area and a warm front with precipitation passed it on 18 January, which inhibited the retrieval of temperature profiles for much of the day (Fig. 3a,b). The warm south-westerly flow gradually descended on 17 and 18 January, decreasing the cold pool depth (*Decay* period). The gradual warming in lower layers and cooling in upper layers eventually eroded the cold pool by 19 January. Adler et al. (2021) found that strong winds and warm air first occurred on the slopes in the southern part of the basin and concluded that mountain waves and downslope winds were involved in the decay.

For the model evaluation, we concentrated on two periods, namely the four-day CAP period and the two-day Decay period. Figures 3c-h depict the temperature and wind speed bias respectively for CTL, EXP, and v4fp1 using domain d02 as an example. To compute the biases, 21 hours of model data (hours 3-24) from each reforecast were used. This resulted in discontinuities in time when the model data shifted from one reforecast run to the next, because each reforecast is initialized independently in a cold-start configuration. During the CAP period, all runs had a persistent warm temperature bias of several °C up to around 2500 m MSL except for the lowest few hundred meters (Fig. 3c,e,g). Looking at the simulated horizontal temperature distribution above ridge height (not shown), we found a general increase of temperature from east to west in the area. Since the warm bias already existed at initialization, we assume that it was not related to the simulated physical processes in the cold pool. Interestingly, the model runs did not exhibit any warm biases relative to the operational radio soundings at Salem in the west close to the Pacific coast or at Spokane in the far north-eastern part of the basin (yellow marker in Fig. 1), probably because the radiosonde data at these stations were assimilated by the RAP (which provided the initial conditions for these



reforecasts). At the beginning of the Decay period the warm bias reached as high as 8 °C just below mean ridge height. Before
the CAP period, especially on 10 January, and after the Decay period on 19 January, the model temperature biases were much
smaller (< 2 °C). All runs generally showed a positive wind speed bias up to around 800 m MSL and a negative bias above up
to the mean ridge height during the CAP period at Wasco (Fig. 3d,f,h). The bias on 17 January, i.e. the first day of the Decay
period, was variable below the mean ridge height and generally positive above. On the second day, a mainly positive bias was
visible below mean ridge height in the d02 v4fp1 run. All runs showed a strong positive wind speed bias (> 5 m s$^{-1}$) for all
heights on 19 Jan, which could possibly be related to an initialization error on that day.

For a complete comparison of temperature bias at Wasco and wind speed bias at all seven radar wind profiler sites within
the Columbia River Basin for all six runs, we computed mean biases averaged over the CAP and Decay periods (Fig. 4). Due
to missing temperature profiles and missing model output on 18 January, the mean bias for the Decay period was computed for
17 January only. The temperature bias showed a continuous but slight improvement (i) when changing the model version from
CTL via EXP to v4 while keeping the horizontal grid spacing the same and (ii) when decreasing the horizontal grid spacing,
i.e. from d01 to d02, for each model version (Fig. 4a). This is consistent with the results of Arthur et al. (2022) who found a
better representation of the CAP with decreased horizontal grid spacing. The warm bias below the mean ridge height during the
CAP period resulted from an erroneous representation of the vertical temperature structure in all runs (Fig. 5b). The observed
near-isothermal layer up to around mean ridge height was missing and the temperature inversion started too close to the surface
in the model resembling the typical structure of a cloud-free cold pool with a surface-based inversion and a smooth decrease
of stability with height (e.g. Whiteman et al., 2001).

The sign of the bias and the bias improvement for newer model versions (as well as for the runs with finer horizontal grid
spacing) was less clear and consistent for wind speed than for temperature and varied for the different stations (Fig. 4b). This
was partly due to gaps in the observational wind speed data (Fig. 3b) which leads to different sample sizes at different sites. As
the sign of the bias at Wasco was mostly positive on 18 January and more variable on 17 January (Fig. 3h), especially below
ridge height, we would expect a clearer signal in the mean wind bias on 18 January. However, due to the missing model output
on that day for some runs, the mean profiles for the Decay period in Fig. 4b are only computed for 17 January, which might
explain the absence of clearer improvements. During the CAP period, the lowest altitude site (Boardman) had a weak positive
bias below approximately 1000 m MSL for all runs, while Yakima and Walla Walla had a slightly negative bias in the same
layer. Wasco and Goldendale, two sites which were fairly close together and located on the slopes in the southern part of the
basin (Fig. 1), both showed mainly positive biases which were strongest below around 800 m MSL. Little improvement in bias
with finer grid spacing or newer model version was found for the three lowest stations, while the positive bias at Wasco and
Goldendale was reduced. Prineville, the highest altitude site in the south of the CRB, had a strong positive bias in all d01 runs
(> 5 m s$^{-1}$). During the Decay period, all sites have a positive wind speed bias, especially below the mean ridge height. The
wind speed bias below the mean ridge height was generally improved with the EXP and v4fp1 versions at all stations, most
clearly seen at Walla Walla and Prineville. This agrees with the findings by Olson et al. (2019b) who identified a reduction
of wind speed bias in the upper part of the cold pool during the decay phase of a different cold pool case due to the model
developments in EXP. Even though the model changes led to a reduction in wind speed bias, all versions still showed a positive



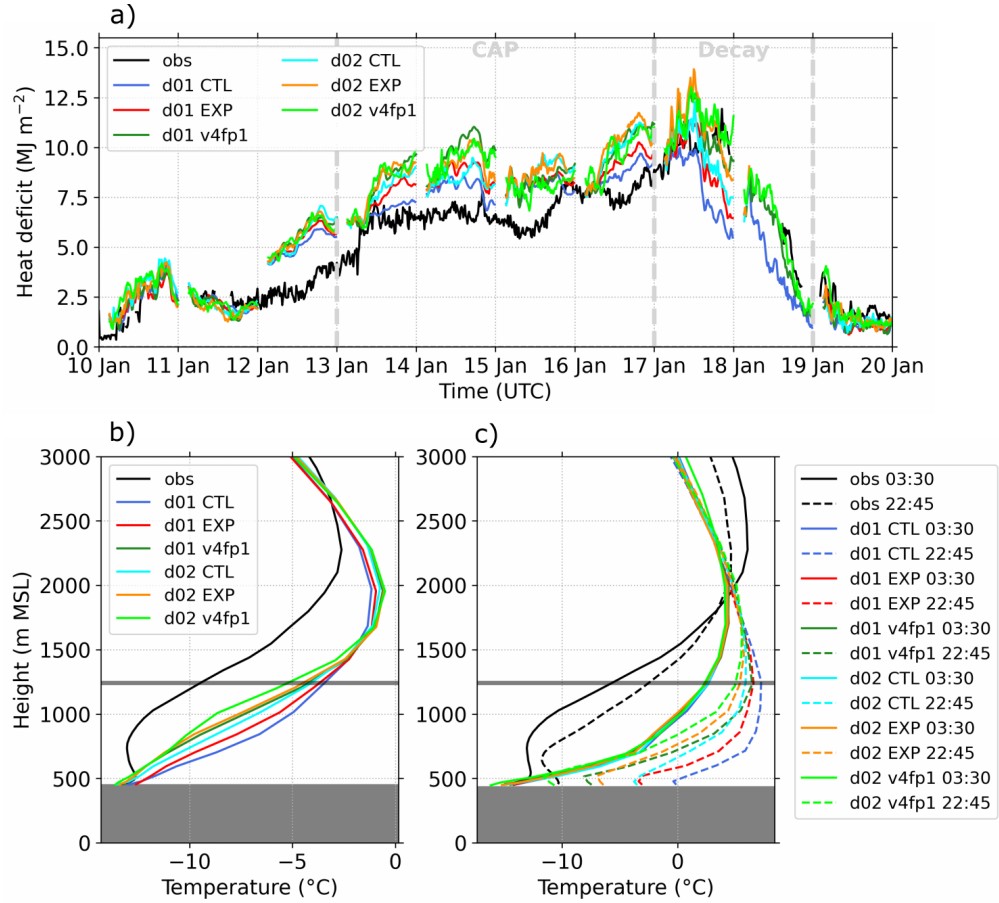

**Figure 5.** (a) Time series of heat deficit (Eq. 1) and profiles of (b) temperature averaged for the CAP period (13 to 16 January), and (c) temperature during the Decay period at 03:30 UTC (solid) and 22:45 UTC (dashed) on 17 January at Wasco from observations and the different model runs. The horizontal grey line indicates the mean ridge height and dark grey shading station height.

bias during the Decay period. The positive wind speed and temperature bias indicate that the warm and strong south-westerly flow still descended too fast into the Columbia River Basin, at least on the first day of the decay.

## 3.2 Cold pool strength and decay

As a proxy for cold pool strength, the heat deficit $Q$ from the surface $h_{sfc}$ up to the mean ridge height $h_{ridge}$ can be used (Whiteman et al., 1999):

$$Q = c_p \int_{h_{sfc}}^{h_{ridge}} \rho(z)[\theta_{h_{ridge}} - \theta(z)] \, dz \tag{1}$$



where $c_p$ is the specific heat capacity of air at constant pressure, $\rho(z)$ is the air density profile, $\theta_{h_{ridge}}$ is the potential temperature at mean ridge height, and $\theta(z)$ denotes the potential temperature profile. Figure 5a shows the heat deficit at Wasco for the whole 10-day period computed from observations and model output. With the warming at and above mean ridge height starting on 12 January (Fig. 3a) the heat deficit increased in the observations. This increase was captured by all model versions likely due to the large-scale character of this warming which was equal in all versions and not impacted by the changes in model physics. All model runs overestimated the heat deficit during the CAP period, which is related to the erroneous high value of the ridge height temperature (Fig. 5b).

When the warm air descended into the basin on 17 January, the cold pool depth and the associated heat deficit decreased (Fig. 5a). The simulated magnitude of the decrease through the end of 17 January, that is until the end of the 24 h forecast period, is stronger than observed with the strongest decrease found in d01 CTL. This decrease in heat deficit can easily be understood when comparing temperature profiles at the beginning and end of 17 January (Fig. 5c). The descending warm air results in observed warming of up to 4 °C above around 750 m and daytime convective heating causes warming near the surface of up to 5 °C. The simulated temperature profiles are nearly identical in all runs shortly after initialization at 03:30 UTC. The impact of the model developments on the cold pool erosion becomes evident in the following 20 h until 22:45 UTC. Connected to the descending warm air, temperature increases in all runs. Even though all runs were too warm compared to the observations, the warming was stronger and extended much further downwards in CTL (d01 and d02) and EXP (d01) resulting in a near-surface temperature increase of nearly 15 °C in d01 CTL and a substantial weakening of the cold pool, which was well reflected in the low heat deficit values at the end of 17 January (Fig. 5a). Similar to 17 January, the heat deficit decreases on 18 January in the available runs, that is d01 CTL, d01 v4fp1, and d02 v4fp1. Albeit still a little too strong compared to the observations, the heat deficit decrease in the v4fp1 runs agreed with the observations much better than the CTL run. Even though we found a good agreement in the observed and simulated temporal evolution of the heat deficit in the improved model version, there was still a high bias in wind speed present at many stations (Fig. 4b) indicating the cold pool depth decreased too quickly.

### 3.3 Dependence of near-surface temperature bias on station height

In Sects. 3.1 and 3.2, we analyzed the vertical temperature structure of the cold pool using profile measurements at one site in the basin. In the following, we evaluate the dependence of the near-surface temperature bias on station height using the large number of stations in the basin available from the MesoWest repository (station locations in Fig. 1). The stations cover a height range up to 2000 m and we computed pseudo-vertical profiles by averaging the absolute values and biases at surface stations falling within 100 m height bins. By doing this any information on horizontal variability was lost, but we found a clear dependence of the bias on station height and less dependence on the location of the station in the basin during the CAP period, which supports the validity of this approach.

Figure 6a shows the temporal evolution of the observed pseudo-vertical temperature profiles during the 10-day period. The cold pool was well visible with cold temperatures at the stations below the mean ridge height and warmer air at stations higher up. A clear diurnal cycle with higher temperatures during the day was evident during the CAP period. The warming during the Decay period first occurred at higher-altitude stations and progressively affected stations at lower altitudes. The near-surface




**Figure 6.** Time height section of (a) observed near-surface temperature and of the temperature bias between d02 (b) CTL, (c) EXP, and (d) v4fp1 runs and the observed values using data from more than 500 stations distributed in the Columbia River Basin (locations in Fig. Fig. 1). Values are averaged over 100 m height bins. The horizontal grey line indicates the mean ridge height and light grey shading missing data. The periods when the cold pool was persistent (CAP) and when it decayed (Decay) are indicated by the vertical dashed lines.




**Figure 7.** 24-h composites of (a) near-surface temperature bias, (b) shortwave downward radiation flux, and (c) longwave downward radiation flux averaged over the CAP period (13-16 January). Before calculating the composites, temperature bias and radiation fluxes are computed at the location of the individual surface stations and then averaged over 100 m height bins. The horizontal dashed lines indicate different height levels with colors corresponding to terrain contours in Fig. 8.

temperature bias was most pronounced during the CAP and Decay periods (Fig. 6b-d). During the CAP period, the biases
are very similar from one day to the other, which makes it suitable to compute 24-h composites by averaging the biases for




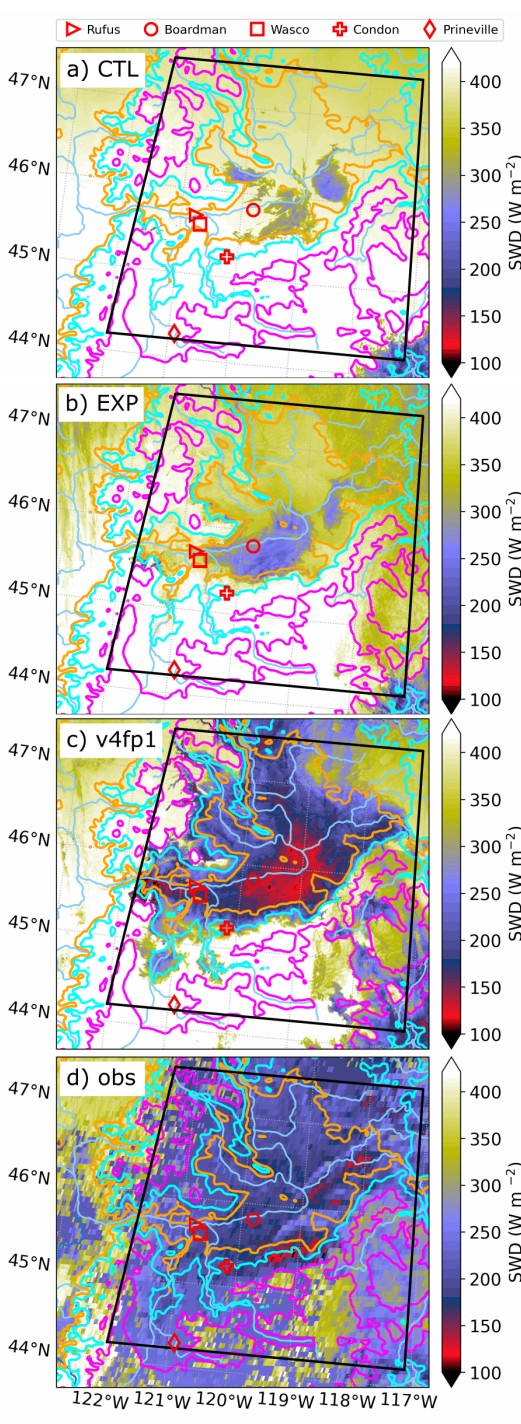

**Figure 8.** Spatial distribution of shortwave downward radiation flux, SWD, in the d02 (a) CTL, (b) EXP, and (c) v4fp1 runs, and (d) detected by satellite at 21 UTC on 14 January. Terrain height contours are indicated by orange (500 m), cyan (750 m), and magenta (1250 m) isolines and rivers are shown in blue. The red markers indicate the locations of stations with radiation flux measurements.





**Figure 9.** Boxplot of (a) daytime shortwave downward radiation bias ΔSWD as well as (b) daytime and (c) nighttime longwave downward radiation bias ΔLWD between the different model runs (versions: CTL, EXP, v4fp1, domains: d01, d02) and the observations at Rufus, Boardman, Wasco, Condon, and Prineville during the CAP period. The white circles indicate the mean biases, boxes show the interquartile range with the median indicated by the horizontal line, and the whiskers extend to the points that lie within 1.5 times the interquartile range of the lower and upper quartiles.





the four days during the CAP period (Fig. 7a). The bias changed its sign with height and roughly three height layers can be distinguished: 0-500 m MSL (Layer 1), 500-1250 m MSL (Layer 2), and above 1250 m MSL (Layer 3) with in general negative biases in Layer 1 and 3 and positive bias in Layer 2. In CTL, the negative bias reached values as low as -8 °C in Layer 3 and -5 °C in Layer 1 during nighttime. A diurnal cycle of the bias was visible in all layers in CTL and somewhat reduced in EXP,

but it was much smaller in v4fp1 (Fig. 6b-d, Fig. 7a). This diurnal cycle was most pronounced in Layer 1 and Layer 2, with a daytime bias regularly exceeding 4 °C and even going up to 8 °C for the d01 CTL run (Fig. 6b, Fig. 7a).

We suspect that the differences in the temperature bias between the different runs were related to differences in low-level cloud cover. Because LWP which contains both resolved grid-scale and subgrid-scale cloud water mixing ratio is only output in the v4 runs, we used downward radiation fluxes as a proxy for clouds. During daytime, clouds lead to a reduction in the

shortwave downward radiation flux. The longwave downward radiation flux can be used during day and night. Since thermal emission at cloud base primarily contributes to this downward longwave flux, it is generally larger in the presence of low clouds compared to cloud free conditions when the troposphere is more transparent in the infrared. Shortwave downward radiation flux increased with station height (Fig. 7b), which indicates that stations at lower altitudes experienced more clouds than stations higher up. While this height dependence was generally visible in all runs, it was most pronounced in both v4fp1 runs. This

means that more clouds were present in the basin, especially at stations below around 750 m MSL, in v4fp1 compared to CTL and EXP. This is well visible in the spatial distribution of shortwave downward radiation flux shown at 21 UTC on 14 January as an example in Fig. 8a-c. Mid and high-level clouds were mostly absent at this time. The area with reduced shortwave downward radiation flux indicating low-level clouds was confined to a much smaller region in the lower part of the basin in CTL (Fig. 8a) and EXP (Fig. 8b) compared to v4fp1 (Fig. 8c). From all runs, the extent of reduced shortwave downward radiation fluxes

in the v4fp1 run agreed best with the observed values (Fig. 8d), especially over terrain lower than 750 m. Over the higher terrain, all runs, even v4fp1, overestimated shortwave downward radiation fluxes at the surface (i.e. they underestimated cloud coverage).

The height dependence of longwave downward radiation flux draws a similar picture (Fig. 7c). Higher values in the EXP and v4fp1 runs, mainly at stations below around 750 m MSL, indicate that more clouds were present on the average compared

to the CTL run during both day and nighttime. Thus, the very cold nocturnal biases in CTL (Fig. 7a) can be explained by fewer clouds which fostered strong radiative cooling, while more clouds reduced the negative bias in EXP and v4fp1. A lack of clouds during daytime allowed too strong radiative heating of the surface in CTL compared to the observations leading to the high positive temperature bias. The warm bias we see in EXP and v4fp1 in Layer 2 might be related to the warm bias we see in the profiles at Wasco (Fig. 3c,e,g, 4a, 5b). Too strong radiative cooling due to the lack of clouds may explain why the

positive nighttime bias in Layer 2 in EXP and v4fp1 was not present in the CTL runs, but was counteracted by cooling.

While Whiteman and Hoch (2014) and Adler et al. (2021) found that pseudo-vertical profiles can be suitable proxies for the free-air temperature structure using observations, this seems not always to be the case in the model. While the warm bias in Layer 2 (Fig. 6a) was consistent with the warm bias in the free-air temperature profiles, at least in some versions (Fig. 3c,e,g), the negative bias in near-surface temperature in Layer 3 is the opposite. This means that the near-surface temperature was

much lower than the free-air temperature in the model (not shown), while they were similar in the observations (Adler et al.,





2021, their Fig. 6). One possible explanation for this is that the model had too few clouds above stations in Layer 3, even in v4fp1. Satellite observations analyzed by Adler et al. (2021) (their Fig. 7) and shown in the example in Fig. 8 indicate that clouds also existed over the higher terrain during the CAP period, although less extensive than at lower terrain, whereas the high shortwave downward and low longwave downward radiation fluxes in the model in Layer 3 point towards very few 330 clouds in that layer (Fig. 7b,c). Another reason might be that the model has problems with correctly representing the near-surface temperature at higher altitudes where the vegetation category changes from mainly savannas, grasslands, and wetlands to forest and potentially the need for a canopy model which is currently not implemented in HRRR. Note that the ground was snow covered in the simulations and in reality.

We have seen that the differences in radiation fluxes between the different model runs are consistent with the differences 335 in temperature bias. The lowest temperature biases in v4fp1 suggest that the downward radiation fluxes in this version agreed best with the observations. To investigate this, we computed shortwave and longwave downward radiation biases at five sites - Rufus, Boardman, and Wasco which are all located in Layer 1 and Condon and Prineville located in the upper part of Layer 2 (Fig. 9). Since the bias depends on the time of the day, we distinguished between day and night and computed the shortwave downward radiation bias for daytime hours (Fig. 9a) and the longwave downward radiation bias for both (Fig. 9b,c). During 340 the day, CTL and EXP had slightly positive biases in shortwave downward radiation flux (Fig. 9a) and consistent with that a negative bias in longwave downward radiation flux (Fig. 9b), indicating a lack of clouds. The sign of these biases changed in v4fp1 at stations in Layer 1, most obvious at Rufus and Wasco, indicating too many or too thick clouds. During nighttime, CTL and EXP had negative longwave downward radiation biases at all sites on the average, again indicating a lack of clouds (Fig. 9c). An improvement is visible in v4fp1, where the mean and median biases got smaller especially at Rufus and Wasco. 345 Little differences are visible at Prineville indicating that clouds did not change much at that site between the different model runs. Combining the findings from shortwave and longwave downward radiation biases, we conclude that v4fp1 represented clouds in Layer 1 better during the nighttime (reduced longwave downward radiation bias), but had too many or too thick clouds in Layer 1 during daytime (negative shortwave downward radiation bias and positive downward radiation bias).

## 4   Cold pool characteristics for a 48-h forecast horizon in v4

In Sect. 3, we have seen that model developments between the CTL, EXP, and v4 versions lead to a better representation of the cold pool structure and low-level clouds with a reduced bias in temperature profiles and near-surface temperature during the first 24 hours of the forecast. Now we extend the analysis to 48 h to investigate if and why the cold pool characteristics change for longer forecast hours and we take a closer look at the dominant factors taking advantage of the comprehensive output in v4. For the definition of the runs see Fig. 2.

### 355  4.1   Impact of clouds on temperature profiles during the CAP period

The comparison between the mean temperature bias at Wasco during the CAP period shows a substantial reduction of the positive temperature bias below around 1500 m MSL from v4fp1 to v4fp2 (Fig. 10b). The mean temperature profiles for v4fp2




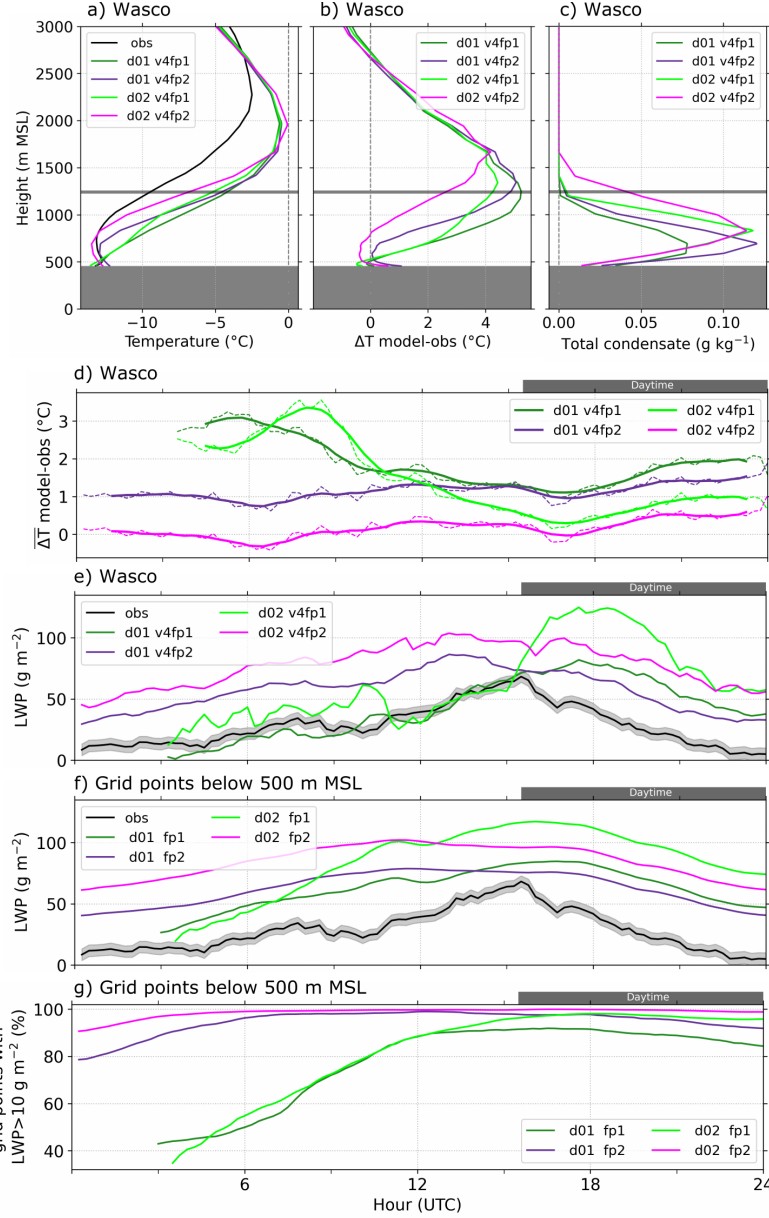

**Figure 10.** Mean profiles of (a) observed and simulated temperature, (b) temperature bias, and (c) simulated total condensate (cloud liquid water, ice, and snow mixing ratio) at Wasco during the CAP period for v4fp1 and v4fp2 runs. The horizontal grey line indicates the mean ridge height and dark grey shading station height. Composite time series of (d) temperature bias averaged up to the mean ridge height and (e) observed and simulated LWP at Wasco; (f) observed LWP at Wasco and simulated mean LWP averaged over all grid points in the Columbia River Basin; and (g) percentage of grid points in the basin with LWP > 10 g m$^{-2}$ during the CAP period. In (d), dashed lines show the 15-min output and solid lines are smoothed with a 2-h window. Shadings in (d) and (e) indicate the 1-$\sigma$ uncertainty of LWP from the TROPoe retrievals. In (e) and (f), only grid points with a terrain height of less than 500 m MSL are used. For the definition of the model runs see Fig. 2.



reveal a near-isothermal layer in the lower few hundred meters, which agrees very well with the observations (Fig. 10a). On average, low-level clouds were present mainly up 1250 m MSL in the simulations, somewhat higher in d02 v4fp2 (Fig. 10c).

This is the same layer in which strongest changes in temperature bias occurred between both forecast periods (Fig. 10a,b). To illustrate the change of temperature bias with time of the day, we compute 24-h composites of the bias over the four days of the CAP period and average them up the mean ridge height, that is the layer in which differences were most pronounced (Fig. 10d). Before around 10:00 UTC, the height-averaged temperature bias at Wasco was several °C larger in v4fp1 than in v4fp2 (Fig. 10d). This means that the bias was largest during the first 10 forecast hours and decreased with longer forecast times

(forecast hours 24-34). Between 10:00 and 24:00 UTC, the bias was more similar for both forecast periods. Because mid- and high-level clouds were largely absent during the CAP period and most of the total condensate was liquid, LWP very well represents the evolution of the low-level clouds and we compute 24-h composites of LWP for the four model runs as well as from observations at Wasco (Fig. 10e). The observed mean LWP at Wasco had a clear diurnal cycle with lower values during the night and maximum values at 15:30 UTC, that is at around sunrise (Fig. 10e). During the course of the daytime period,

the observed mean LWP decreased and nearly diminished at sunset. Clear differences are visible in the simulated temporal evolution of LWP for the two forecast periods (Fig. 10e), even though the total condensate profiles have similar maximum values when averaged over time, especially for d02 runs (Fig. 10c). More than 60 % of the simulated LWP originated from the resolved gridscale clouds on average. LWP in v4fp1 had a pronounced diurnal cycle with low values at the beginning and maximum values at around 18 UTC. After 18 UTC, that is during the afternoon, LWP decreased like in the observations.

The simulated decrease was however weaker than observed, which resulted in higher LWP values at sunset at around 24:00 UTC compared to the observations and means that LWP at the beginning of v4fp2 was too high. The overestimation of LWP persisted for the whole 24-h period of v4fp2. The temporal evolution of LWP can explain the differences in temperature bias between both forecast periods (Fig. 10b). Because clouds were usually missing at model initialization time during the CAP period and only slowly formed with time (Fig. 10e,f), the mean temperature structure in v4fp1 (Fig. 10a) resembled a typical

cloud-free cold pool with the inversion starting at the surface (e.g. Whiteman et al., 2001). When the clouds thickened and LWP increased (Fig. 10e) they modified the temperature profiles making them more similar to the observed profiles (Fig. 10a) and the temperature bias decreased (Fig. 10d).

To investigate the representativeness of the results at Wasco, we averaged LWP over all grid points in the basin with a terrain height of less than 500 m MSL (Fig. 10f). Inspection of the spatial distribution of LWP (not shown) revealed that the maximum

spatial extent of low-level clouds roughly follows the 500-m terrain contour. Thus, the average over these grid points gives a good estimate of the low-level clouds in the basin. The general temporal behavior of LWP was similar to that at Wasco. That is (i) LWP was low at the beginning of v4fp1 and increased during the first 15 h after initialization, (ii) the model underestimated the decrease of LWP during daytime, and (iii) the underestimated decrease of LWP in v4fp1 during the daytime (i.e., from 15 to 24 UTC) led to high LWP in v4fp2. To analyze how extensive the low-level cloud cover in the basin was, we computed

the percentage of grid points in the basin with LWP$> 10$ g m$^{-1}$ (Fig. 10g). At the beginning of v4fp1, less than 50 % of all grid points in the basin were cloudy. During the subsequent hours, clouds became more widespread, reaching nearly hundred




percent in d02 at around 18 UTC. During v4fp2, nearly all grid points in the basin were cloudy during the whole 24 h, especially in d02.

The small impact of forecast length on temperature and clouds during daytime could be related to the initialization time of
the runs at 00:00 UTC. We have seen that cloud cover and LWP increases in the hours after initialization, that is during the night, which results in similar values of LWP during daytime (Fig. 10e,f). If the runs were initialized at 12:00 UTC instead, it is likely that differences in clouds between the two forecast periods would be largest during daytime and less pronounced during nighttime, since the model had more time to generate clouds. The investigation of this is however beyond the scope of this study. A lack of clouds at initialization, e.g. if they are not inherited at initialization, can lead to unrealistic drops
in temperature and the formation of fog layers over snow covered surfaces at high latitudes and the failure of the model to correctly forecast low-level clouds (Hagman et al., 2021). However, since low-level clouds formed in the cold pool in the Columbia River Basin, albeit slowly, this seems not to be an issue here.

### 4.2 Dependence of cold pool strength and decay on forecast period

In Sect. 3.2, we have seen that all model runs captured the timing of the decay of the cold pool on 17 and 18 January fairly
well 24 h in advance, although the erosion was too strong, even in the newest model version. Here we investigate how well the newest model version v4 forecasts the cold pool evolution and decay up to 48 h in advance. Figure 11a shows the heat deficit at Wasco computed with Eq. 1 from the observations and from the v4fp1 and v4fp2 runs. While the overall temporal evolution was well represented in v4fp2, some differences are evident. At the beginning of the decay period on 17 January, the heat deficit was overestimated in v4fp2 (Fig. 11a). However, the heat deficit in v4fp2 decreased slightly faster than in v4fp1
which resulted in very similar heat deficit values at the end of the day. The differences in heat deficit at the beginning of 17 January arose from the different temperature profiles (solid lines in Fig. 11b). As a result of the low-level clouds present in the cold pool in v4fp2 (Sect. 4.1), a near-isothermal layer topped by an elevated inversion was present in v4fp2, which was absent in v4fp1. Until the end of the day, descending warm air results in the formation of a strong temperature inversion up to around 1000 m MSL which was very similar in both forecast periods (dashed lines in Fig. 11b). Differences in the vertical structure
were even larger between runs with different horizontal grid spacing than they were between different forecast periods. This means that despite the differences in stratification at the beginning of the Decay period, the degree of erosion at the end of the first day was very similar in both forecast periods. This suggests that large-scale forcing was the decisive factor for the cold pool decay and that the pre-decay stratification did not matter in this case.

On 18 Jan, observed and simulated heat deficits for both forecast periods were very similar (Fig. 11a) in agreement with
similar temperature profiles (Fig. 11c). However, the heat deficit in all runs was slightly lower than observed at the end of this day, which was reflected in warmer temperatures near the surface and indicates a too fast erosion. Arthur et al. (2022) found that using a 3D PBL scheme instead of the 1D PBL scheme better maintained the temperature structure in the lower few hundred meters on 18 January.



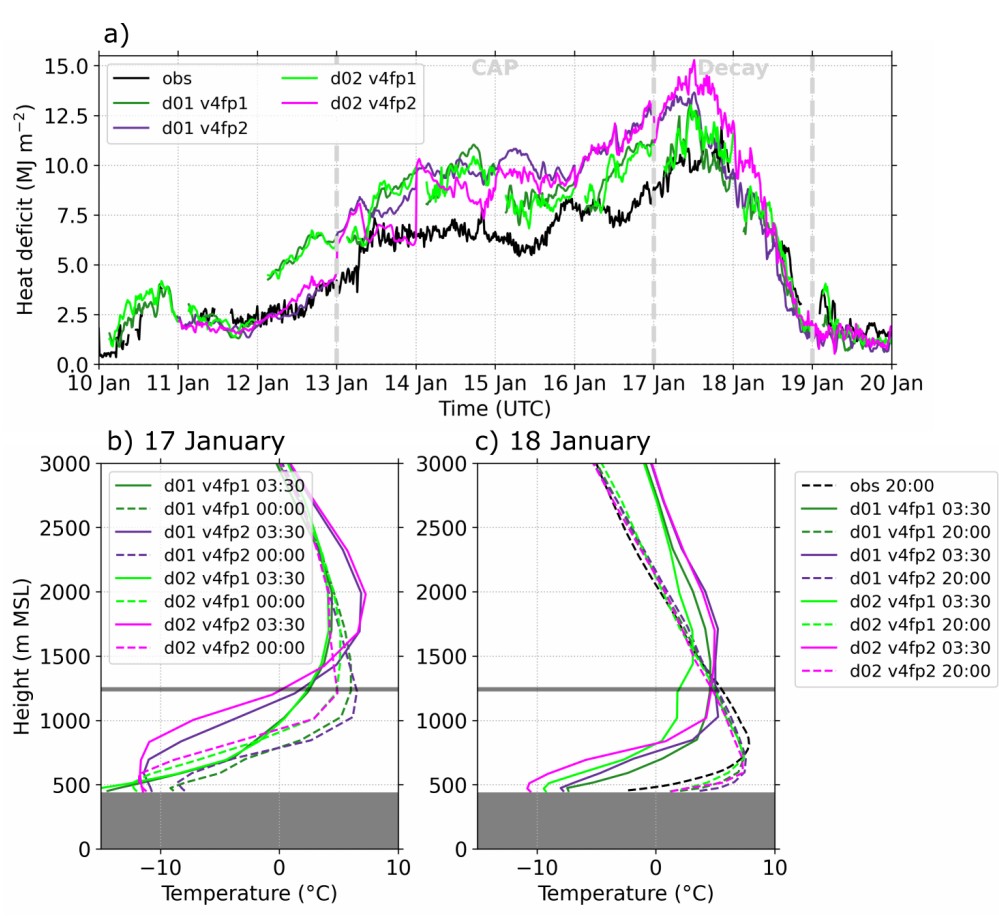

**Figure 11.** (a) Time series of heat deficit (Eq. 1) from observations and model runs v4fp1 and v4fp2 at Wasco. Profiles of simulated temperature at Wasco during the Decay period on (b) 17 January and (c) on 18 January at the beginning (solid) and end (dashed) of the respective day. In (c), the observed temperature profile at 20:00 UTC is added. The horizontal grey line indicates the mean ridge height and dark grey shading station height.



### 4.3 Impact of clouds on the near-surface temperature bias during the CAP period

At surface stations below around 500 m MSL (Layer 1), a cold temperature bias was visible during the CAP period in v4fp1 (Fig. 7a). This cold bias turned into a warm bias in v4fp2, especially during nighttime (Fig. 12a). The change in temperature bias between v4fp1 and v4fp2 was small at stations above 500 m MSL. To understand if this change in the sign of the bias at stations below 500 m MSL during the night was related to the more extensive clouds during v4fp2 below 500 m MSL (Fig. 10f,g), we compared the change in temperature to the change in LWP between the two forecast periods for stations below

500 m. Figure 12b,c shows the relationship between LWP and temperature differences for individual stations for all nighttime hours during the four day CAP period. At the majority of stations which detected an increase in temperature from v4fp1 to v4fp2, LWP also increased (first quadrant in Fig. 12b,c). The relationship between LWP changes and temperature changes is in particular evident for stations and times when LWP in v4fp1 was small (red contours in Fig. 12b,c). If LWP was already large ($> 50$ g m$^{-2}$) in v4fp1, temperature did not change much between the two forecast periods, probably because the clouds were

already opaque in the longwave and thus were contributing maximum downward radiation flux in v4fp1. This demonstrates the warming effect low-level clouds have at the surface during nighttime.

The differences in clouds in the lower part of the basin between both forecast periods are consistent with biases in shortwave and longwave downward radiation fluxes during the CAP period (Fig. 13). During daytime, shortwave downward radiation bias was negative and longwave downward radiation bias was positive on average at stations within Layer 1 with differences

between both forecast periods being small (Fig. 13a,b), implying that the model had too many clouds regardless of the forecast period. During nighttime, the negative longwave downward radiation bias at Rufus and Boardman in v4fp1 was nearly eliminated in v4fp2, while at Wasco a positive bias appeared which was not present in v4fp1 (Fig. 13c). This can be explained by a spatial variability in the observed fluxes (and thus clouds) which was not correctly represented in the simulations. The mean observed nighttime longwave downward radiation flux was higher at Rufus and Boardman than at Wasco indicating more

clouds at the lower altitude stations. While the simulated flux at Wasco in v4fp1 agreed fairly well with the observations on average, it was much too low at Boardman meaning that the model did not capture the intense clouds at Boardman. In v4fp2, the simulated longwave downward radiation flux was high at both sites on average, indicating similar cloud cover in contradiction to the observations. The higher fluxes in v4fp2 better agreed with the observations at Rufus and Boardman, thus reducing the bias, while they overestimated the fluxes at Wasco, causing the positive bias. The warm bias near the surface at stations below

500 m (Fig. 12a) together with the positive bias longwave downward radiation flux (Fig. 13c) and the too high LWP (Fig. 10e) indicate that the extent and thickness of low-level clouds in the basin during nighttime in v4fp2 was too large.

### 5 Summary and conclusions

In this study, we evaluated three different versions of NOAA's HRRR model for two horizontal grid spacings ($\Delta x$=750 m and $\Delta x$= 3 km) for a persistent cold pool event in the Columbia River Basin. For the evaluation we used remote sensing and

in situ observations of temperature, wind, and radiation fluxes gathered during the WFIP2 field campaign and near-surface temperature measurements from a large number of stations downloaded from the MesoWest repository. A key component of





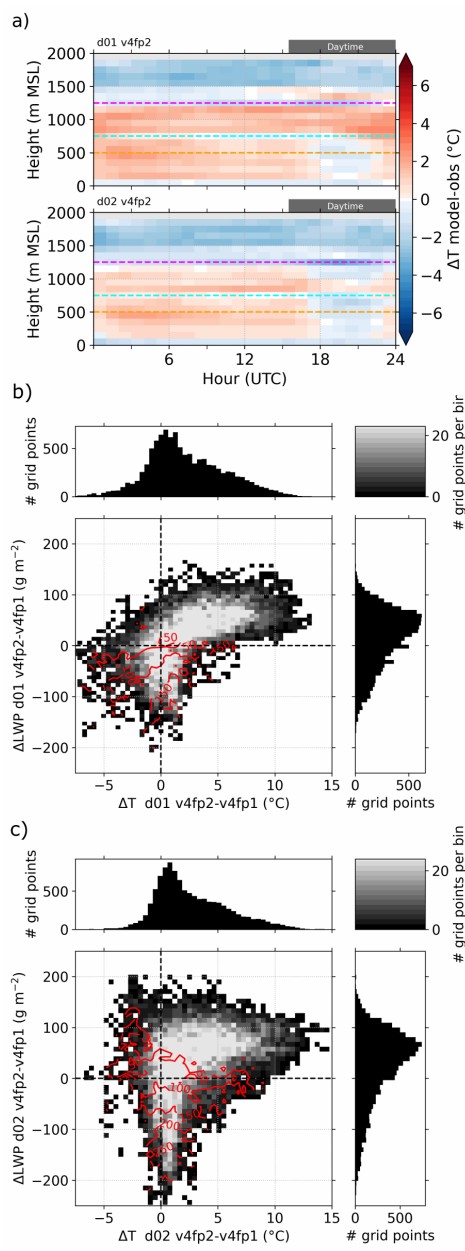

**Figure 12.** (a) 24-h composites of near-surface temperature bias averaged over the CAP period (13-16 January) for the v4fp2 run. Before calculating the composites, temperature bias and radiation fluxes are computed at the location of the individual surface stations and then averaged over 100 m height bins. The horizontal dashed lines indicate different height levels with colors corresponding to terrain contours in Fig. 8. Relationship of the change in liquid water path (ΔLWP) and near-surface temperature (ΔT) between both forecast periods (v4fp2-v4fp1) for (b) domain d01 and (c) domain d02. The distributions are computed for surface stations below 500 m MSL for all nighttime hours during the CAP period. Red contours indicate the distribution of LWP in v4fp1.






**Figure 13.** Boxplot of (a) daytime shortwave downward radiation bias ΔSWD as well as (b) daytime and (c) nighttime longwave downward radiation bias ΔLWD between different model runs (versions: v4fp1, v4fp2, domains: d01, d02) and the observations at Rufus, Boardman, Wasco, Condon, and Prineville during the CAP period. The white circles indicate the mean biases, boxes show the interquartile range with the median indicated by the horizontal line, and the whiskers extend to the points that lie within 1.5 times the interquartile range of the lower and upper quartiles.





the WFIP2 project were model developments in HRRR to improve the forecast for wind energy applications. The three HRRR versions under evaluation were the version which was run operationally at the beginning of WFIP2 (CTL), a version which uses model developments targeted in WFIP2 (EXP), and a version very close to the currently operational HRRR version (v4fp1 and v4fp2). Our aim was to (i) investigate how changes in the model physical parameterizations and numerical methods impacted the cold pool structure and evolution, in particular temperature and low-level clouds, and (ii) if and how the model performance changed for a longer reforecast horizon.

In a first step, we compared the three different versions and two different horizontal grid spacings during the 4-day persistent CAP period and the 2-day Decay period. For this we used 24-h reforecasts initialized at 00:00 UTC.

- Low-level clouds were observed during the CAP period which were associated with a near-isothermal sub-cloud layer at a profiling site at Wasco in the basin. In all model simulations, the mean vertical temperature profiles at this site resembled the typical structure of a cloud-free cold pool with a surface-based inversion indicating that clouds were underestimated either in frequency or thickness. This led to a mostly warm bias. During the cold pool decay, high temperature and high wind speed biases were present indicating a too fast top-down erosion of the cold pool. Although present in all versions, the biases were reduced in EXP and v4fp1 compared to CTL. Similarly, the biases were also reduced when the model used smaller horizontal grid spacing (finer resolution).

- The near-surface temperature bias at more than 500 stations in the Columbia River Basin area showed a clear dependence on station height. In contrast to the positive temperature bias through most of the lowest 2.5 km in the free atmosphere, stations below 500 m MSL and above the mean ridge height at 1250 m MSL showed a negative bias and stations in between often a positive bias. A strong diurnal cycle of the temperature bias occurred in the CTL runs, which was somewhat reduced in the EXP runs and was nearly eliminated in the v4fp1 runs. The differences in surface temperature bias between the different model versions were consistent with differences in longwave and shortwave downward radiation fluxes which we used as a proxy for low-level clouds. Nocturnal longwave downward radiation biases were smallest in v4fp1 pointing to more realistic clouds during the night. On the other hand, the radiation biases indicated an overestimation of clouds during daytime in the newest model version.

In a second step, we investigated the performance of the newest model version (v4 runs) for longer forecast hours by dividing the 48-h long reforecasts in half and comparing both forecast periods (v4fp1 and v4fp2).

- Cloud characteristics were fairly similar during the daytime for both forecast periods, while more clouds were present during the night in v4fp2. The difference during the night arose because few clouds were present at model initialization at 00:00 UTC (around sunset) and clouds gradually formed during the first 15 hours of the forecast, i.e. during much of the nighttime period of v4fp1, while clouds were already present at sunset and persisted through the night in v4fp2. During daytime, simulated LWP was high during both forecast periods and decreased less than observed, indicating insufficient clearing of the clouds. The warm bias in the temperature profiles below mean ridge height was much reduced in the lower 500 m during the CAP period in v4fp2, due to the increased presence of low-level clouds and a near-isothermal sub-cloud layer.



– Despite the differences in clouds and temperature during the CAP period between both forecast periods, the timing of the cold pool decay was equally well captured in v4fp2. This indicates that for this cold pool event large-scale forcing was decisive for its decay and that the pre-decay stratification did not matter.

– The cold bias at surface stations below 500 m present in v4fp1 during nighttime turned into a warm bias in v4fp2, while differences during daytime were small. This change in near-surface temperature bias during nighttime was related to an increase in LWP in v4fp2 at stations which had few clouds (low LWP) during v4fp1. Consistent with the differences in temperature bias between both forecast periods, differences in daytime shortwave downward radiation bias were small while a positive longwave downward radiation bias occurred during nighttime for v4fp2 at some stations related to a too extensive cloud cover in the simulations.

In short, we found that the model development efforts during WFIP2 and subsequent refinements included in v4 greatly improved the simulation of the cold pool characteristics, in particular of temperature and clouds. While all model versions were lagging low-level clouds at initialization, clouds gradually formed with time and were most realistic in the newest model version. The improved representation of clouds resulted in a reduced near-surface temperature and radiation bias and a more realistic vertical temperature structure. However, clouds did not clear sufficiently during daytime leading to an overestimation of clouds during the day and for longer forecast hours which introduced a warm bias near the surface during the second night

of the reforecast.

In this study, we evaluated the model developments as a whole, that is we did not investigate the sensitivity to the individual changes in physical parameterizations and numerical methods. Based on our findings and the findings by Olson et al. (2019b) and Berg et al. (2021), we assume that changes in mixing length, computation of horizontal diffusion in Cartesian space, 6th

order filter, subgrid-scale clouds, and small-scale gravity wave drag had an impact while the changes to the mass-flux scheme of the MYNN-EDMF parameterization were less relevant. We found evidence that the overestimation of clouds for longer forecast hours (v4fp2) is related to the failure of the model to sufficiently clear clouds during daytime. Wilson and Fovell (2018) tested model changes to the Weather Research and Forecasting (WRF) model to better forecast radiative cold pools and fog in California's Central Valley and found that adding a new entrainment term which is controlled by the fluxes at the PBL

top and which allowed entrainment to be generated at cloud top by radiative and evaporative cooling to the Yonsei University (YSU) PBL scheme helped to lift and dissolve fog layers. In the MYNN-EDMF scheme used in the v4 simulations no turbulent mixing at cloud top was present, which could have entrained dry air and helped to erode the more than 500 m deep cloud layer. Model developments similar to the Wilson and Fovell (2018) approach are currently under development for the HRRR model. It is thus likely that the reduced diffusion in the PBL scheme, horizontal diffusion, and 6th order filter in v4 combine to now

undermix in stable conditions, especially without the cloud-top cooling mechanism.

*Code and data availability.* All WFIP2 observational data and CTL and EXP model data are available on the DOE Data Archive and Portal (https://a2e.energy.gov/data#wfip2). v4 model data and plotting scripts to produce the figures in this manuscript are available at Zenodo via



https://doi.org/10.5281/zenodo.6713495. The model code used for the CTL and EXP runs is available via https://doi.org/10.5281/zenodo.3369984 and the code used for the v4 runs is available via https://doi.org/10.5281/zenodo.6672455.

*Author contributions.* BA completed the data analysis and prepared the manuscript with contributions from JMW, LB, IVD, JBO, and DDT. JK conducted the model simulations with HRRR version 4 and IVD assisted with the CTL and EXP model outputs.

*Competing interests.* The authors have no competing interests to declare.

*Acknowledgements.* We thank all of the individuals for help with the WFIP2 site selection, leases, instrument deployment and maintenance, data collection, and data quality control. Funding for this work was provided by the U.S. Department of Energy (DOE) Office of Energy
Efficiency and Renewable Energy, Wind Energy Technologies Office, and by the NOAA Atmospheric Science for Renewable Energy program. This work was supported by the NOAA Cooperative Agreement with CIRES, NA17OAR4320101 and the NOAA Physical Sciences Laboratory and was made possible in part because of the data made available by the governmental agencies, commercial firms, and educational institutions participating in MesoWest. The development of GSIP algorithms and associated products has been supported by the NOAA/NESDIS GOES Product Systems Development and Implementation (G-PSDI) program (Donald Gray, G-PSDI program manager,
and Tom Schott, satellite product manager).



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
