# Peer review of "Evaluation of a cloudy cold-air pool in the Columbia River Basin in different versions of the HRRR model"

_EGUsphere, 2022_

## Referee Comment (RC1)

**General Comments**

This manuscript evaluates the development, evolution, and dissipation of a cold-air pool (CAP) in different versions of the HRRR model. Specifically, they consider HRRR v1 (CTL), a modified version of CTL that used model improvements from the WFIP2 field campaign (EXP), and HRRR v4 (v4fp1) that included several modifications including to the MYNN PBL scheme. Each of these was applied to two different grid spacings. The model data were compared against in situ and remote observations in and around the Columbia River Basin. The paper is generally well-written and the topic is relevant to EGU readers. The authors motivate the need for improved model representation of, e.g., winds in the considered conditions due to the reduced efficiencies in wind turbines. I have a few issues with the manuscript as presented.

First, I wonder why the authors use a cold-start approach starting at exactly 00 UTC with a 3-hour spin-up? The current method results in 3-hour periods of empty data in, e.g., Fig. 3 (which the authors note). Is this a limitation of the RAP output used to supply the initial conditions? If not, did the authors consider an approach where they initialize the model at 21 UTC on the day prior, use the same 3-hour spin-up, then run for 24-hours from 00–00 UTC? Even if RAP necessitated the 00 UTC initialization, why not initialize at the same 00 UTC and run for 24 hours from 03–03 UTC and avoid such gaps? Or was this all because the CTL and EXP were run this way in the past and the authors did not want to duplicate efforts? It seems like an unnecessary constraint. Second, and related, did the authors consider running (at least the v4fp1) a simulation for an entire CAP duration. I understand computational limitations and am only curious.

Also in terms of the computational setup, I wondered if the 3-hour spin-up was adequate for the considered cases? I would think there might be a substantial response and adjustment to the topography and land-use in the region at the considered scales when starting from scratch. Don't the results at the end when 48-hour runs were considered at least give some credence to this idea? To that end, which land-surface model was used and what is the terrain resolution? Research has also shown that long spin-up periods in the LSM are needed for improved land-surface representation. Given the focus on fluxes at the surface, those seem like relevant considerations, especially when the inner domain has a spacing of 750 m. That leads to questions about the PBL scheme. Was there any consideration to how running at 750 m might affect the PBL representation of a diurnal cycle given that this scale range is likely within the well-known grey zone of atmospheric turbulence?

Lastly, I would have expected more quantitative assessments of performance since this is a purported evaluation. For instance, many of the comparisons are presented as 2D color plots, which can be misleading. To be clear, I have no doubt v4fp1 was overall a better tool based on the authors' work, but I think some areas need more care. For instance, Section 3.2 introduces heat deficit as a proxy for cold pool strength, which is shown in Fig. 5 as a time trace. The positive traits of v4fp1 in the initial creation and decay periods were discussed at length in this section, but the middle portions to my eye show several periods where CTL and EXP were closer to the observations and yet the

text merely said "All model runs overestimated the heat deficit during the CAP period." I think it is reasonable to expect more explicit statistical analysis of this CAP period as it relates to the heat deficit and other fields given the title of the paper. As it stands, a lot of the evaluation is in the form of plots and many potential analyses are lost in, e.g., daily composites. Box plots are limited to fluxes, which are used as a proxy for clouds. Even then, there are plenty of locations where the mean biases in CTL and EXP are closer or as-close as in the v4fp1 cases. The results seem site dependent, which the authors address, but more discussion could be had as to how misses in certain periods or layers could result in better matches during other periods or in other locations (i.e., the so-called idea of getting the right answer for the wrong reason). That can all be related back to the issues above in terms of more discussion about the LSM, terrain, spin-up periods, grid spacing, etc.

Based on the above considerations, I believe this paper requires enough work that it would look substantially different than it does in its present form. In addition to these broad issues, I have a few specific issues that are listed below. Accordingly, I recommend that the manuscript require **major revisions** before it is suitable for publication in *EGUsphere*.

**Specific Comments**

Line 28  The authors reduced "cold-air pool" to "cold pool" on line 15, but use the full version here. Check for consistency.

Fig. 5  In print, these line colors were hard to delineate—especially the green ones. Is it possible to consider dotted lines for d02 domains of the same color as their d01 counterparts?

Fig. 6  There is an extra "Fig" in the caption text on line 2.

Line 275  Suggest changing "agreement in" to "agreement between."

Line 292  Consider rewording "with in general negative…"

---

## Referee Comment (RC2)

Review of: **Evaluation of a cloudy cold-air pool in the Columbia River Basin in different versions of the HRRR model**

This paper evaluates the performance of a suite of WRF-based simulations of cloud cold-air pools in the Columbia River basin. The study is motivated by a need to improve NWP for wind energy, especially during variations in stratification and windiness in the boundary layer associated with cold-air pools. Model performance, spanning variations in physics and resolution, are evaluated using WFIP2 observations. Changes in physics include variations if formulation of the mixing length, inclusion of mass-flux terms for non-local mixing, subgrid cloud processes, gravity wave drag, a wind farm parameterization, and differences in diffusion and filtering. Model results are examine for two forecast periods. Results indicate some improvement in developing low level clouds, yielding better agreement in near surface temperatures during the first simulation period, but also produced some errors in cloudiness and temperatures in the later parts of the event/simulations. This mixed result make the take aways from this paper somewhat unclear, though there is obviously a strong radiative impact of clouds on CAP structure and evolution that must be simulated correctly to get the correct forecast. The paper includes many modifications to the model between the different experiments, but not in a well-defined "sensitivity" study approach that would allow a detailed assessment of individual changes. This is understandable in that the authors are working with quasi-operational model codes, rather than a suite of idealized simulations. Nevertheless the well-constructed observation-to-model comparisons provide some physical and numerical insights into how to improve CAP simulations with respect to clouds, and offers some tentative insights into what future work is needed to further improve cloud process representation (e.g., entrainment mixing). I find the paper incrementally advances our understanding of and ability to simulate CAPs, and I do not find any fatal flaws in the approach or results. I therefore believe the paper should be published following addressing a handful of specific concerns detailed below.

Major points, the first substantive, the second editorial

(1) There needs to be some improved discussion of observation and retrieval uncertainties. These include: (a) potential biases in the TROPoe retrievals due to smoothing of the profile and (b) LWP estimates. This is important since there are some large temperature biases and the reader needs to have a basis for understanding the comparative magnitude of simulation biases and observational biases. Perhaps these points are addressed in some of the referenced literature, but some discussion of these points needs to be surfaced.

(2) Conciseness: I'd encourage the authors to try to trim as many words as possible from the manuscript. There is a lot decipher so trimming extra verbiage would be helpful. I have a few specific examples below.

Specific Comments (more major comments in **bold**):

Abstract. I had to read the abstract a number of times, having stumbled over the wording and meaning. A few examples:

    a. The phase "also related to this" strikes me as odd.

    b. "Differences between different model versions were in particular visible" -> could just be "Differences amongst model versions were apparent in simulated temperature and near-surface clouds"

    c.

Line 49: "the challenges" -> "these challenges". More broadly this tough to read sentence, consider revising if possible.

Line 55: "Besides the conduction" -> "besides completing"? conduction implies a physical process (e.g., heat transfer)

Line 93: "adapted" -> do you mean adopted?

Line 148: "A wind farm parameterization" -> probably needs a citation here so we know the details of this. Also, I don't recall seeing any further discussion of this modification in the results, so is it really necessary to detail this? As it is I was expecting to see something about this later on.

Line 172-176: This is one location you could explain if there are expected biases or uncertainties in the observations/retrievals that would impact our interpretation of the model-obs biases discussed throughout.

Lines 183-202: This paragraph is long and could be cleaned up to make it easier to interpret. There are many opportunities to rephrase and reduce the word count in here. Just one minor example on line 196: "from the easterly direction" could just be "easterly".

Line 189-: "exhibited" rather than "experienced"?

**Lines 207-220**: The notable warm bias is evaluated against the observations, but there is no discussion here as to potential bias in the observations either here on lines 100-110, where TROPoe was discussed. How good are the TROPoe profiles compared against other data sources? Could some of the bias here be from issues with retrievals from the observations? From the observations themselves (e.g., RASS). Perhaps these are addressed in Adler et al. 2021… but I did not have time to dig into that.  The TROPoe profiles look quite smooth… are they washing out sharp layers that then results in parts of the profile that are too warm while others are too cold?

Line 220: "could be related to an initialization error on this day"… these are the sorts of things that make it tricky to take clear messages away from this paper. There are a mish-mash of things causing the observed discrepancies. No fault of the authors, but diminishes the impact of these analyses.

Line 227-231: Redundant with my comment above, but do you have any radiosonde observations at this time to compare with the TROPoe profile and the model profiles. The TROPoe profile strikes me as getting some of the critical details but also likely smoothing the capping inversion structure… which could results in some of the warm bias in the model (though the model is obviously getting the low levels totally wrong)

Section 3.3: In general I really like this section, finding the analysis nicely done and the results relatively easy to interpret. One suggestion is to move some of the discourse on Lines 321-330 to the intro of this section so that the reader is better equipped to think about the expected differences between pseudo-profiles and free-air profiles (e.g., impact of near surface heating/cooling on station data as compared to free air data).

Figure 10. panels a,bc,d,e all have the same "Wasco" heading. This is somewhat confusing/not helpful. Maybe either give them a topical name a.) profiles, b.) biases, c) condensate profiles…. etc or just use a), b), c) with no descriptor.

**Line 368**: Again we are in need of knowing how well the radiometer observation measure the LWP. The model-observation analysis requires that we have at least some sense for how uncertain the observations are here. How does this LWP compare with radiosonde observations?

Line 446: strike "intense" and either remove the superlative or use a more physical descriptor (high cloud fraction, deep louds, large LWP, etc)

Line 453: "evaluated three different versions" -> "evaluated three versions"

**Line 465**: I'm again wondering if the "near isothermal" structure is an artifact of the TROPoe, whereas an actual sounding would show something close to moist adiabatic? Should we conclude that the subcloud layer is truly characterized by "near isothermal" conditions in the CAPs?

Line 491: "was equally well capture in v4fp2" and what? The control V2fp1? The other runs?

Line 500: "greatly improve" -> I'd remove this superlative here and just say "improved"

Line 502: Should "lagging" be "lacking"?

Line 502: Most "realistic"-> perhaps indicate how so? Better cloud fraction, better diurnal cycle, etc?

Line 503: I might make this point clearer in the introduction as well. This study is not a clear sensitivity study that specifically isolates one process at a time. Setting the readers expectation on this early on helps to frame the paper.

Lines 513-518. These are important points about the lack of entrainment mixing in EDMF for stratocumulus type cloud that you are making here. I wonder if you could introduce a few of these thoughts when discussing the model results for fp2 above.

---

## Author Comment (AC1)

**Response to reviewer comments**

Manuscript: egusphere-2022-355
Title: Evaluation of a cloudy cold-air pool in the Columbia River Basin in different versions of the HRRR model
Authors: Bianca Adler, James M. Wilczak, Jaymes Kenyon, Laura Bianco, Irina V. Djalalova, Joseph B. Olson, and David D. Turner

We thank the two anonymous reviewers for their clear and helpful comments, which helped to improve the manuscript. In the following we provide a point-to-point response to all reviewer comments. The reviewers' comments are printed in italic and our response in roman font type. We indicate the line numbers of the revised manuscript where revisions have been made. For the reviewers' convenience we also copied larger changes we made to the manuscript to this response and enclosed them with quotation marks.

**Reviewer 1**

**General comments**

1. *This manuscript evaluates the development, evolution, and dissipation of a cold-air pool (CAP) in different versions of the HRRR model. Specifically, they consider HRRR v1 (CTL), a modified version of CTL that used model improvements from the WFIP2 field campaign (EXP), and HRRR v4 (v4fp1) that included several modifications including to the MYNN PBL scheme. Each of these was applied to two different grid spacings. The model data were compared against in situ and remote observations in and around the Columbia River Basin. The paper is generally well-written and the topic is relevant to EGU readers. The authors motivate the need for improved model representation of, e.g., winds in the considered conditions due to the reduced efficiencies in wind turbines. I have a few issues with the manuscript as presented.*

**Response:** Thank you for your suggestions.

2. *First, I wonder why the authors use a cold-start approach starting at exactly 00 UTC with a 3-hour spin-up? The current method results in 3-hour periods of empty data in, e.g., Fig. 3 (which the authors note). Is this a limitation of the RAP output used to supply the initial conditions? If not, did the authors consider an approach where they initialize the model at 21 UTC on the day prior, use the same 3-hour spin-up, then run for 24-hours from 00–00 UTC? Even if RAP necessitated the 00 UTC initialization, why not initialize at the same 00 UTC and run for 24 hours from 03–03 UTC and avoid such gaps? Or was this all because the CTL and EXP were run this way in the past and the authors did not want to duplicate efforts? It seems like an unnecessary constraint. Second, and related, did the authors consider running (at least the v4fp1) a simulation for an entire CAP duration. I understand computational limitations and am only curious.*

**Response:** The CTL and EXP runs were not done specifically for this study, but were part of the WFIP2 model development efforts. The strategy was to run these tests in a similar (short-range) configuration to the operational RAP/HRRR (without the data assimilation). This is why the forecast horizon was 24 hr. For consistency, we performed the v4 simulations in exactly the same manner as the CTL and EXP runs (just extended to 48 hr). Because we consider the first 3 hours as spin up, the gaps are unavoidable when comparing the three model versions. We agree that this is not ideal, but we also think that a different configuration (like the reviewer suggested) would not drastically change the results of our study.

Since our goal was to compare v4 to the previous CTL and EXP runs, we chose the same configuration for v4 to be consistent and did not perform a single simulation for the entire cool period.

We added this sentence (l. 171):

'The v4 version was run specifically for this study with the same setup as CTL and EXP for consistency...'

3. *Also in terms of the computational setup, I wondered if the 3-hour spin-up was adequate for the considered cases? I would think there might be a substantial response and adjustment to the topography and land-use in the region at the considered scales when starting from scratch. Don't the results at the end when 48-hour runs were considered at least give some credence to this idea? To that end, which land-surface model was used and what is the terrain resolution? Research has also shown that long spin-up periods in the LSM are needed for improved land-surface representation. Given the focus on fluxes at the surface, those seem like relevant considerations, especially when the inner domain has a spacing of 750 m. That leads to questions about the PBL scheme. Was there any consideration to how running at 750 m might affect the PBL representation of a diurnal cycle given that this scale range is likely within the well-known grey zone of atmospheric turbulence?*

**Response:** The HRRR runs were initialized off of the RAP model, which is designed for short-range forecasting. Benjamin et al. (2016) shows that the skill of the RAP is typically higher at hr 0 and hr 1 and goes down after. This does depend on the phenomenon of interest and may also vary in location, but it generally holds true. Because the RAP is designed for short-range forecasting, it has a tighter fit to the observations than a data assimilation system used for medium-range forecasting like the Global Data Assimilation system (GDAS) which is used for the GFS (Benjamin et al. 2016). For this reason, initializing off of the GFS likely would require more spin up time than initializing off of the RAP. Spinup timescales are still an open science question. For the WFIP2 runs, spinup problems became difficult to spot after forecast hour 3. This is why d02 was initialized at this forecast hour and we disregarded the first 3 hours for the d01 runs.

We agree with the reviewer that soil state consistency is very important regarding spin-up time. The best performing LSM is usually the one that is initialized off of a model that runs

the same or very similar LSM. One recent example is Min et al. 2022, who discuss soil state consistency and how that probably determined the outcome of which LSM performed the best. The LSM used in the HRRR runs in our study and in the RAP is the RUC LSM (Benjamin et al. 2016, Dowell et al. 2022). This means that soil state consistency is achieved in our runs and that no further spin-up is required.

We added the following sentence to the manuscript (l. 164):

'This relatively short spin-up time is possible because the RAP is designed for short-range forecasting and its data assimilation system has a tighter fit to observations than a data assimilation system used for medium-range forecasting (Benjamin et al., 2016). Furthermore, soil state consistency is achieved by using the same land surface model in the RAP and in the HRRR (Dowell et al., 2022).'

The 750 m horizontal resolution of d02 was specifically chosen for the WFIP2 model development to test and develop the MYNN-EDMF in a (convective boundary layer) grey zone configuration. Some of the research done to make the MYNN-EDMF work better within the greyzone was published in Angevine et al. (2020). Lessons learned in that work were applied to the v4 configurations used in this study. However, in stable conditions (like they are present in the cold pool), all turbulent motions are sub-750 m (horizontal resolution of d02) in scale. This means that for the phenomena studied here, this is not really a grey zone problem. In addition we found that the MYNN-EDMF was largely inactive during the cold pool.

4. *Lastly, I would have expected more quantitative assessments of performance since this is a purported evaluation. For instance, many of the comparisons are presented as 2D color plots, which can be misleading. To be clear, I have no doubt v4fp1 was overall a better tool based on the authors' work, but I think some areas need more care. For instance, Section 3.2 introduces heat deficit as a proxy for cold pool strength, which is shown in Fig. 5 as a time trace. The positive traits of v4fp1 in the initial creation and decay periods were discussed at length in this section, but the middle portions to my eye show several periods where CTL and EXP were closer to the observations and yet the text merely said "All model runs overestimated the heat deficit during the CAP period." I think it is reasonable to expect more explicit statistical analysis of this CAP period as it relates to the heat deficit and other fields given the title of the paper. As it stands, a lot of the evaluation is in the form of plots and many potential analyses are lost in, e.g., daily composites. Box plots are limited to fluxes, which are used as a proxy for clouds. Even then, there are plenty of locations where the mean biases in CTL and EXP are closer or as-close as in the v4fp1 cases. The results seem site dependent, which the authors address, but more discussion could be had as to how misses in certain periods or layers could result in better matches during other periods or in other locations (i.e., the so-called idea of getting the right answer for the wrong reason). That can all be related back to the issues above in terms of more discussion about the LSM, terrain, spin-up periods, grid spacing, etc.*

**Response:** For a more quantitative assessment, we modified Fig. 3 and added a table summarizing temperature and wind biases. We believe this makes it much easier to quantify the model improvements. Thank you for this suggestion. The model improvements were hard to see in the time height sections of the temperature and wind speed biases. We now only show the temperature and wind bias between the CTL version and the observations. For EXP and v4fp1, we show the change in bias instead. This shows very nicely that biases were strongly improved below ridge height during the CAP and Decay period. The table includes mean biases up to mean ridge height for the three versions during the CAP and Decay period for temperature at Wasco and wind averaged over all 7 sites.

This is the new Fig. 3 and Table 1:

[Figure]

**Figure 3.** Time height section of observed (a) temperature profiles (color-coded) and potential temperature (isolines) and (b) horizontal wind speed (color-coded) and horizontal wind vector (arrows) at Wasco. Time height section of (c) temperature, $T$, bias and (d) horizontal wind speed, $U$, bias for d01 CTL and the changes in (e,g) temperature biases and (f,h) horizontal wind speed biases between EXP and CTL and v4fp1 and CTL, respectively, at Wasco. The green dots show observed cloud base height in (a) and the green contours show the simulated 0.1 g kg$^{-1}$ isoline of total condensate (cloud water, snow and ice mixing ratio) in (g). The horizontal grey line indicates the mean ridge height, dark grey shading station height, and light grey shading missing data. The CAP and Decay periods are indicated by the vertical dashed lines.

**Table 2.** Mean temperature and wind speed biases during the CAP and Decay periods averaged up to the mean ridge height. The wind speed biases are an average over all seven stations with wind profile measurements.

| HRRR run | Temperature bias at Wasco (°C) | | Wind speed bias averaged over all sites (m s$^{-1}$) | |
|---|---|---|---|---|
| | CAP | Decay | CAP | Decay |
| CTL d01 | 3.7 | 8.5 | 1.1 | 2.6 |
| EXP d01 | 3.1 | 7.0 | 1.0 | 2.0 |
| v4fp1 d01 | 2.0 | 5.4 | 1.1 | 2.2 |
| CTL d02 | 2.4 | 6.4 | 0.5 | 1.6 |
| EXP d02 | 1.9 | 4.8 | 0.5 | 1.4 |
| v4fp1 d02 | 1.5 | 4.4 | 0.6 | 1.4 |

The heat deficit in all runs is overestimated because of a warm bias near ridge height already present at initialization (this is described in l. 221ff). The heat deficit in the CTL run is smallest and thus closest to the observation, However, this is not because the thermal stratification in the cold pool is better captured, but because the profiles are warmest below ridge height and agree worst with the observations. This means that the heat deficit agrees best for CTL, but for the wrong reason. We added a sentence in l. 277:

'Heat deficit in CTL d01 was smallest and thus closest to the observations, but this was not because thermal stratification was most accurate in this run. It was rather caused by the curvature of the temperature profile below ridge height in CTL d01 (Fig. 5b).'

5. *Based on the above considerations, I believe this paper requires enough work that it would look substantially different than it does in its present form. In addition to these broad issues, I have a few specific issues that are listed below. Accordingly, I recommend that the manuscript require major revisions before it is suitable for publication in EGUsphere.*

**Response:** We hope we sufficiently address your issues in our revised version.

**Specific Comments**
1. *Line 28 The authors reduced "cold-air pool" to "cold pool" on line 15, but use the full version here. Check for consistency.*

**Response:** Changed.

2. *Fig. 5 In print, these line colors were hard to delineate—especially the green ones. Is it possible to consider dotted lines for d02 domains of the same color as their d01 counterparts?*

**Response:** To make the lines easier to distinguish, we followed the reviewer's suggestion and now use the same color for each version and solid and dashed line style for d01 and d02,

respectively. To better accommodate color blind readers, we also replaced green with orange (v4).

This is the new figure 5:

[Figure]

**Figure 5.** (a) Time series of heat deficit (Eq. 1) and profiles of (b) temperature averaged for the CAP period (13 to 16 January), and (c) temperature during the Decay period at 03:30 UTC (thin lines) and 22:45 UTC (thick lines) on 17 January at Wasco from observations and the different model runs. The horizontal grey line indicates the mean ridge height and dark grey shading station height.

3. *Fig. 6 There is an extra "Fig" in the caption text on line 2.*

**Response:** Changed.

4. *Line 275 Suggest changing "agreement in" to "agreement between."*

**Response:** Changed.

**Reviewer 2**

This paper evaluates the performance of a suite of WRF-based simulations of cloud cold-air pools in the Columbia River basin. The study is motivated by a need to improve NWP for wind energy, especially during variations in stratification and windiness in the boundary layer associated with cold-air pools. Model performance, spanning variations in physics and resolution, are evaluated using WFIP2 observations. Changes in physics include variations if formulation of the mixing length, inclusion of mass-flux terms for non-local mixing, subgrid cloud processes, gravity wave drag, a wind farm parameterization, and differences in diffusion and filtering. Model results are examine for two forecast periods. Results indicate some improvement in developing low level clouds, yielding better agreement in near surface temperatures during the first simulation period, but also produced some errors in cloudiness and temperatures in the later parts of the event/simulations. This mixed result make the take aways from this paper somewhat unclear, though there is obviously a strong radiative impact of clouds on CAP structure and evolution that must be simulated correctly to get the correct forecast. The paper includes many modifications to the model between the different experiments, but not in a well-defined "sensitivity" study approach that would allow a detailed assessment of individual changes. This is understandable in that the authors are working with quasi-operational model codes, rather than a suite of idealized simulations. Nevertheless the well-constructed observation-to-model comparisons provide some physical and numerical insights into how to improve CAP simulations with respect to clouds, and offers some tentative insights into what future work is needed to further improve cloud process representation (e.g., entrainment mixing). I find the paper incrementally advances our understanding of and ability to simulate CAPs, and I do not find any fatal flaws in the approach or results. I therefore believe the paper should be published following addressing a handful of specific concerns detailed below.

**Response:** Thank you for your suggestions.

**Major points, the first substantive, the second editorial**

1.  *There needs to be some improved discussion of observation and retrieval uncertainties. These include: (a) potential biases in the TROPoe retrievals due to smoothing of the profile and (b) LWP estimates. This is important since there are some large temperature biases and the reader needs to have a basis for understanding the comparative magnitude of simulation biases and observational biases. Perhaps these points are addressed in some of the referenced literature, but some discussion of these points needs to be surfaced.*

**Response:** We added several sentences at various locations in the manuscript to specify the retrieval uncertainty and related it to the magnitude of the model biases. Because TROPoe uses an optimal estimation framework, it outputs a posterior covariance matrix as a measure of the uncertainty of the retrieval. This includes contributions from uncertainties in the observations, the prior, and the forward model (e.g. Turner and Loehnert 2021). The 1-sigma uncertainties of the temperature profiles (that is the diagonal elements of the posterior covariance matrix) in our study were less than 1.5 °C in the lowest 2.5 km, which

are typical values for this instrument combination (Djalalova et al. 2022). The 1-sigma uncertainty of retrieved LWP was around 5 kg/m2. While these uncertainties are not small, they are smaller than the temperature biases found between the model and the observations at distinct levels (Fig. 3 and 4a). Adler et al. 2021, compared the retrieved temperature profiles to pseudo-vertical profiles computed from a large number of surface stations in the area and found a good agreement in depth and strength of the temperature inversion. Pseudo-vertical profiles can be good proxies for free-air temperature within a few degrees especially during wintertime (Whiteman and Hoch, 2014).

We added information on the TROPoe uncertainty in l. 110:

'Since TROPoe uses the optimal estimation framework, it outputs a posterior covariance matrix as a measure of the uncertainty of the retrieval, which includes contributions from uncertainties in the observations, the prior, and the forward model (e.g., Turner and Löhnert, 2021). The 1-sigma uncertainties of the temperature profiles (that is the diagonal elements of the covariance matrix) in our study were less than 1.5 °C in the lowest 2.5 km, which are typical values for this instrument combination (Djalalova et al., 2022). The 1-sigma uncertainty of retrieved LWP was around 5 kg/m2.

We added information why we think the retrieved profiles are realistic in l. 202:

'We are confident in the accuracy of the retrieved temperature profiles, because Adler et al. (2021) found a good agreement in depth and strength of the temperature inversion between the retrieved free-air profiles and pseudo-vertical profiles derived from the many surface stations in the area and because the differences in low-level stratification during cloudy and cloud-free periods are physically consistent.'

We added information on the model bias magnitude with respect to retrieval uncertainty in l. 220:

'The magnitude of the model bias is much larger than the 1-sigma uncertainty of the retrieved temperature profiles (Sect. 2.1).'

2. *Conciseness: I'd encourage the authors to try to trim as many words as possible from the manuscript. There is a lot decipher so trimming extra verbiage would be helpful. I have a few specific examples below.*

**Response:** We tried to trim as many words as possible.

**Specific Comments**

1. *Abstract. I had to read the abstract a number of times, having stumbled over the wording and meaning. A few examples:*
   *a. The phase "also related to this" strikes me as odd.*
   *b. "Differences between different model versions were in particular visible" -> could just be "Differences amongst model versions were apparent in simulated temperature and near-surface clouds"*

**Response:** Changed.

2. *Line 49: "the challenges" -> "these challenges". More broadly this tough to read sentence, consider revising if possible.*

   **Response:** Changed. The new sentence is (l. 48):

   'To improve the forecast for wind energy applications over complex terrain, the Second Wind Forecast Improvement Project (WFIP2) was initiated by the Department of Energy in 2015 (Shaw et al., 2019).'

3. *Line 55: "Besides the conduction" -> "besides completing"? conduction implies a physical process (e.g., heat transfer)*

   **Response:** Changed.

4. *Line 93: "adapted" -> do you mean adopted?*

   **Response:** Changed.

5. *Line 148: "A wind farm parameterization" -> probably needs a citation here so we know the details of this. Also, I don't recall seeing any further discussion of this modification in the results, so is it really necessary to detail this? As it is I was expecting to see something about this later on.*

   **Response:** Details on the wind farm parameterizations are given in Olson et al. 2019. However, because the wind farm parameterization is not relevant in our study, we decided to remove it here.

6. *Line 172-176: This is one location you could explain if there are expected biases or uncertainties in the observations/retrievals that would impact our interpretation of the modelobs biases discussed throughout.*

**Response:** We now discuss retrieval uncertainties and its relevance for the model evaluation in l. 110, 202, and 220.

7. *Lines 183-202: This paragraph is long and could be cleaned up to make it easier to interpret. There are many opportunities to rephrase and reduce the word count in here. Just one minor example on line 196: "from the easterly direction" could just be "easterly".*

**Response:** We put this paragraph into a new section 3.1 'Observed temperature and wind profiles' and divided it into three paragraphs making it easier to read.

8. *Line 189-: "exhibited" rather than "experienced"?*

**Response:** Changed.

9. *Lines 207-220: The notable warm bias is evaluated against the observations, but there is no discussion here as to potential bias in the observations either here on lines 100-110, where TROPoe was discussed. How good are the TROPoe profiles compared against other data sources? Could some of the bias here be from issues with retrievals from the observations? From the observations themselves (e.g., RASS). Perhaps these are addressed in Adler et al. 2021... but I did not have time to dig into that. The TROPoe profiles look quite smooth... are they washing out sharp layers that then results in parts of the profile that are too warm while others are too cold?*

**Response:** The TROPoe profiles are smooth because they are retrieved with an optimal estimation framework. Most of the information comes from brightness temperature measurements from a microwave radiometer, which is a passive remote instrument. The RASS provides data in a layer of a few hundred meters depth in the lower part of the cold pool. This means that most of the information in the inversion layer comes from the microwave radiometer brightness temperature measurements. This is an ill-posed problem as there could exist multiple profiles that would yield the observed brightness temperature. The profiles are unavoidably smoother than profiles from active remote sensing instruments like Raman lidar. While the TROPoe retrievals can struggle to resolve sharp elevated inversion (Djalalova et al. 2022), the retrievals do well in resolving the more than 1000 m deep temperature inversion in our cold pool.
Besides the posterior covariance matrix which provides information on the retrieval uncertainty (see response to major point 1), the retrieval outputs the averaging kernel (Turner and Löhnert 2014). The trace of the averaging kernel provides the degrees of freedom, which is a measure of the number of independent pieces of information from the observation used in the solution. The rows of the averaging kernel provide a measure of the smoothing as a function of height that results from the retrievals and give an estimate of the vertical resolution at each height of the retrieval solution. If the retrieved profiles were to be compared to profiles with high vertical resolution, such as radiosonde profiles, the averaging kernel could be applied to the high-resolution profiles to avoid representativeness errors. However, because we compare the retrieved profiles to model data which are already on a rather coarse vertical grid, we think it is valid to compare the simulated and observed profiles directly.

10. *Line 220: "could be related to an initialization error on this day"... these are the sorts of things that make it tricky to take clear messages away from this paper. There are a mish-*

*mash of things causing the observed discrepancies. No fault of the authors, but diminishes the impact of these analyses.*

**Response:** We agree and we are aware that other factors than parameterizations can impact the model biases. Nevertheless, we think that the improvements we find for the newer version can be attributed to the model developments for horizontal diffusion, filtering, subgrid-scale clouds, and small-scale gravity drag. The improvements clearly occurred in the lower part of the cold pool where these developments were active.

*11. Line 227-231: Redundant with my comment above, but do you have any radiosonde observations at this time to compare with the TROPoe profile and the model profiles. The TROPoe profile strikes me as getting some of the critical details but also likely smoothing the capping inversion structure… which could results in some of the warm bias in the model (though the model is obviously getting the low levels totally wrong)*

**Response:** Unfortunately, no radiosondes were launched in the part of the Columbia River Basin where the WFIP2 instrumentation was installed. Operational radiosondes were launched twice daily in the far northeastern part of the basin at Spokane (location in Fig. 1 in manuscript). We compared the mean temperature profiles from the TROPoe retrievals at Wasco (obs) and from radio soundings at Spokane and found a fairly good agreement in inversion top height and maximum temperature (Response Fig. 1). Differences in low-level temperature structure can probably be attributed to spatial differences in low-level clouds. This along with the good agreement with pseudo-vertical temperature profiles and the fact that changes in stratification were associated with changes in horizontal wind (wind turned from easterly below the inversion to varying direction in the inversion) give confidence that the retrieved temperature structure is realistic. Adler et al. 2021 found strong differences in low-level stability between cloudy and cloud-free periods which were physically consistent, which further increased credibility.

[Figure]

Response Fig. 1: Mean temperature profiles from TROPoe retrievals at Wasco (obs, black line) and radio soundings at Spokane (blue line) averaged for the CAP period.

12. *Section 3.3: In general I really like this section, finding the analysis nicely done and the results relatively easy to interpret. One suggestion is to move some of the discourse on Lines 321-330 to the intro of this section so that the reader is better equipped to think about the expected differences between pseudo-profiles and free-air profiles (e.g., impact of near surface heating/cooling on station data as compared to free air data).*

**Response:** Thank you. We added this sentence to the beginning of the section:

'Pseudo-vertical temperature profiles can be good proxies for free-air temperature within a few degrees during wintertime (Whiteman and Hoch 2014, Adler et al. 2021). In general, the pseudo-vertical profiles are colder during the night and warmer during the day because of surface cooling and heating.

13. *Figure 10. panels a,bc,d,e all have the same "Wasco" heading. This is somewhat confusing/not helpful. Maybe either give them a topical name a.) profiles, b.) biases, c) condensate profiles....etc or just use a), b), c) with no descriptor.*

**Response:** We removed the descriptor.

*14. Line 368: Again we are in need of knowing how well the radiometer observation measure the LWP. The model-observation analysis requires that we have at least some sense for how uncertain the observations are here. How does this LWP compare with radiosonde observations?*

**Response:** The 1-sigma uncertainty of the TROPoe LWP retrievals is around 5 g/m3. This is indicated by the shading in Fig. 10e,f in the manuscript. The differences in observed and simulated LWP is thus much larger than the uncertainty. Unfortunately, there were no radio soundings in the area of the WFIP2 instrumentation and even if there were, it is not possible to compute LWP from radiosonde launches.

*15. Line 446: strike "intense" and either remove the superlative or use a more physical descriptor (high cloud fraction, deep louds, large LWP, etc)*

**Response:** Replaced with 'higher cloud fraction'.

*16. Line 453: "evaluated three different versions" -> "evaluated three versions"*

**Response:** Changed.

*17. Line 465: I'm again wondering if the "near isothermal" structure is an artifact of the TROPoe, whereas an actual sounding would show something close to moist adiabatic? Should we conclude that the subcloud layer is truly characterized by "near isothermal" conditions in the CAPs?*

**Response:** This is a good point. Instead of describing the layer as 'near-isothermal', we now say 'weakly stratified' throughout the manuscript, which is a more accurate description.

*18. Line 491: "was equally well capture in v4fp2" and what? The control V2fp1? The other runs?*

**Response:** 'equally well captured in both forecast periods'. This is added now.

*19. Line 500: "greatly improve" -> I'd remove this superlative here and just say "improved"*

**Response:** Changed.

*20. Line 502: Should "lagging" be "lacking"?*

**Response:** Changed.

*21. Line 502: Most "realistic"-> perhaps indicate how so? Better cloud fraction, better diurnal cycle, etc?*

**Response:** Cloud cover was most realistic in the newest model version. We added this in the manuscript.

*22. Line 503: I might make this point clearer in the introduction as well. This study is not a clear sensitivity study that specifically isolates one process at a time. Setting the readers expectation on this early on helps to frame the paper.*

**Response:** We added this information to the introduction (l. 59):

'Note that we did not investigate the sensitivity to the individual changes in physical parameterization and numerical methods, but evaluated the model developments as a whole.'

*23. Lines 513-518. These are important points about the lack of entrainment mixing in EDMF for stratocumulus type cloud that you are making here. I wonder if you could introduce a few of these thoughts when discussing the model results for fp2 above.*

**Response:** We moved parts of this discussion up to Sect. 4.1 where we discuss the cloud evolution in the model (l. 401):

'The too weak LWP decrease indicates a failure of the model to clear clouds during daytime due to insufficient mixing either by bottom-up convection or top-down mixing at cloud top. In the MYNN-EDMF scheme used in the v4 simulations no turbulent mixing at cloud top was present, which could have entrained dry air and helped to erode the more than 500 m deep cloud layer. Wilson and Fovell (2018) tested model changes to the Weather Research and Forecasting (WRF) model to better forecast radiative cold pools and fog in California's Central Valley and found that adding a new cloud-top entrainment term to the Yonsei University (YSU) PBL scheme helped to lift and dissolve fog layers. This term is controlled by the fluxes at the PBL top and allowed entrainment to be generated at cloud top by radiative and evaporative cooling.'

The discussion in the summary is shortened now (l. 541):

'HRRR v4 does not include a parameterization of entrainment generated at cloud top by radiative and evaporative cooling. This feature is currently under development. It is thus likely that the reduced diffusion in the PBL scheme, horizontal diffusion, and 6th order filter in v4 combine to now undermix in stable conditions, especially without the cloud-top cooling mechanism.'

**References:**

Angevine, W. M., Olson, J., Gristey, J. J., Glenn, I., Feingold, G., & Turner, D. D. (2020). Scale Awareness, Resolved Circulations, and Practical Limits in the MYNN–EDMF Boundary Layer

and Shallow Cumulus Scheme, Monthly Weather Review, 148(11), 4629-4639, https://doi.org/10.1175/MWR-D-20-0066.1.

Benjamin, S. G., Weygandt, S. S., Brown, J. M., Hu, M., Alexander, C. R., Smirnova, T. G., Olson, J. B., James, E. P., Dowell, D. C., Grell, G. A., et al. (2016): A North American hourly assimilation and model forecast cycle: The Rapid Refresh, Mon. Wea. Rev., 144, 1669–1694, https://doi.org/10.1175/MWR-D-15-0242.1.

Dowell, D. C., Alexander, C. R., James, E. P., Weygandt, S. S., Benjamin, S. G., Manikin, G. S., Blake, B. T., Brown, J. M., Olson, J. B., Hu, M., Smirnova, T. G., Ladwig, T., Kenyon, J. S., Ahmadov, R., Turner, D. D., and Alcott, T. I. (2022): The High-Resolution Rapid Refresh (HRRR): An hourly updating convection-permitting forecast model. Part 1: Motivation and system description.,Wea. Forecasting, https://doi.org/https://doi.org/10.1175/WAF-D-21-0151.1.

Min, L., Min, Q., & Du, Y. (2022). Evaluation of model summertime boundary layer cloud development over complex terrain in New York State, Weather and Forecasting (published online ahead of print 2022), https://doi.org/10.1175/WAF-D-21-0172.1.